# Bridging the Gap Between Vision Transformers and Convolutional Neural Networks on Small Datasets

**Zhiying Lu, Hongtao Xie***, **Chuanbin Liu***, **Yongdong Zhang**
University of Science and Technology of China, Hefei, China
arieseirack@mail.ustc.edu.cn, {htxie,liucb92,zhyd73}@ustc.edu.cn

## Abstract

There still remains an extreme performance gap between Vision Transformers (ViTs) and Convolutional Neural Networks (CNNs) when training from scratch on small datasets, which is concluded to the lack of inductive bias. In this paper, we further consider this problem and point out two weaknesses of ViTs in inductive biases, that is, the **spatial relevance** and **diverse channel representation**. First, on spatial aspect, objects are locally compact and relevant, thus fine-grained feature needs to be extracted from a token and its neighbors. While the lack of data hinders ViTs to attend the spatial relevance. Second, on channel aspect, representation exhibits diversity on different channels. But the scarce data can not enable ViTs to learn strong enough representation for accurate recognition. To this end, we propose Dynamic Hybrid Vision Transformer (DHVT) as the solution to enhance the two inductive biases. On spatial aspect, we adopt a hybrid structure, in which convolution is integrated into patch embedding and multi-layer perceptron module, forcing the model to capture the token features as well as their neighboring features. On channel aspect, we introduce a dynamic feature aggregation module in MLP and a brand new "head token" design in multi-head self-attention module to help re-calibrate channel representation and make different channel group representation interacts with each other. The fusion of weak channel representation forms a strong enough representation for classification. With this design, we successfully eliminate the performance gap between CNNs and ViTs, and our DHVT achieves a series of state-of-the-art performance with a lightweight model, 85.68% on CIFAR-100 with 22.8M parameters, 82.3% on ImageNet-1K with 24.0M parameters. Code is available at https://github.com/ArieSeirack/DHVT.

## 1 Introduction

Convolutional Neural Networks (CNNs) have dominated in Computer Vision (CV) field as the backbone for various tasks like classification [1, 2, 3, 4, 5, 6, 7], object detection [8, 9, 10] and segmentation [11, 12, 13]. These years have witnessed the rapid growth of another promising alternative architecture paradigm, Vision Transformers (ViTs). They have already exhibited great performance in common tasks, such as classification [14, 15, 16, 17, 18, 19], object detection [20, 21, 22] and segmentation [23, 24].

ViT [14] is the pioneering model that brings Transformer architecture [25] from Natural Language Processing (NLP) into CV. It has a higher performance upper bound than standard CNNs, while it is at the cost of expensive computation and extremely huge amount of training data. The vanilla ViT needs to be firstly pre-trained on the huge dataset JFT-300M [14] and then fine-tuned on the common dataset ImageNet-1K [26]. Under this experimental setting, it shows higher performance than standard CNNs. However, when training from scratch on ImageNet-1K only, the accuracy is

---

*Corresponding author

36th Conference on Neural Information Processing Systems (NeurIPS 2022).

much lower. From the practical perspective, most of the datasets are even smaller than ImageNet-1K, and not all the researchers can hold the burden of pre-training their own model on large datasets and then fine-tuning on the target small datasets. Thus, an effective architecture for training ViTs from scratch on small datasets is demanded.

Recent works [27, 28, 29] explore the reasons for the difference in data efficiency between ViT and CNNs, and draw a conclusion to the lack of inductive bias. In [27], it points out that *with not enough data, ViT does not learn to attend locally in earlier layers.* And in [28], it says that *the stronger the inductive biases, the stronger the representations. Large datasets tend to help ViT learn strong representations. Locality constraints improve the performance of ViT.* Meanwhile, in recent work [29], it demonstrates that *convolutional constraints can enable strongly sample-efficient training in the small-data regime.* The insufficient training data makes ViT hard to derive the inductive bias of attending locality, thus many recent works strive to introduce local inductive bias by integrating convolution into ViTs [18, 15, 30, 31, 32] and modify it to hierarchical structure [33, 34, 16, 17, 35], making ViTs more like traditional CNNs. This style of hybrid structure shows comparable performance with strong CNNs when training from scratch on medium dataset ImageNet-1K only. But the performance gap on much smaller datasets still remains.

Here, we consider that the scarce training data weakens the inductive biases in ViTs. Two kinds of inductive bias need to be enhanced and better exploited to improve the data efficiency, that is, the **spatial relevance** and **diverse channel representation**. **On spatial aspect**, tokens are relevant and objects are locally compact. The important fine-grained low-level feature needs to be extracted from the token and its neighbors at the earlier layers. Rethinking the feature extraction framework in ViTs, the module for feature representation is the multi-layer perceptron (MLP) and its receptive field can be seen as only itself. So ViTs depend on the multi-head self-attention (MHSA) module to model and capture the relation between tokens. As is pointed out in work [27], with less training data, lower attention layers do not learn to attend locally. In other words, they do not focus on neighboring tokens and aggregate local information in the early stage. As is known, capturing local features in lower layers facilitates the whole representation pipeline. The deep layers sequentially process the low-level texture feature into high-level semantic features for final recognition. Thus ViTs have an extreme performance gap compared with CNNs when training from scratch on small datasets. **On channel aspect**, feature representation exhibits diversity in different channels. And ViT has its own inductive bias that different channel group encodes different feature representation of the object, and the whole token vector forms the representation of the object. As is pointed out in work [28], large datasets tend to help ViT learn strong representation. The insufficient data can not enable ViTs to learn strong enough representation, thus the whole representation is poor for accurate classification.

In this paper, we solve the performance gap of training from scratch on small datasets between CNNs and ViTs and provide a hybrid architecture called Dynamic Hybrid Vision Transformer (DHVT) as a substitute. We first introduce a hybrid model to address the issue **on spatial aspect**. The proposed hybrid model integrates a sequence of convolution layers in the patch embedding stage to eliminate the non-overlapping problem, preserving fine-grained low-level features, and it involves depth-wise convolution [36] in MLP for local feature extraction. In addition, we design two modules for making feature representation stronger to solve the problem **on channel view**. To be specific, in MLP, depth-wise convolution is adopted for the patch tokens, and the class token is identically passed through without any computation. We then leverage the output patch tokens to produce channel weight like Squeeze-Excitation (SE) [4] for the class token. This operation helps re-calibrate each channel for the class token to reinforce its feature representation. Moreover, in order to enhance interaction among different semantic representations of different channel groups and owing to the variable length of the token sequence in vision transformer structure, we devise a brand new token mechanism called "head token". The number of head tokens is the same as the number of attention heads in MHSA. Head tokens are generated by segmenting and projecting input tokens along the channel. The head tokens will be concatenated with all other tokens to pass through the MHSA. Each channel group in the corresponding attention head in the MHSA now is able to interact with others. Though maybe the representation in each channel and channel group is poor for classification on account of insufficient training data, the head tokens help re-calibrate each learned feature pattern and enable a stronger integral representation of the object, which is beneficial to final recognition.

We conduct experiments of training from scratch on various small datasets, the common dataset CIFAR-100, and small domain datasets Clipart, Painting and Sketch from DomainNet [37] to examine the performance of our model. On CIFAR-100, our proposed models show a significant performance

margin with strong CNNs like ResNeXt, DenseNet and Res2Net. The Tiny model achieves 83.54% with only 5.8M parameters, and our Small model reaches the state-of-the-art 85.68% accuracy with only 22.8M parameters, outperforming a series of strong CNNs. Therefore, we eliminate the gap between CNNs and ViTs, providing an alternative architecture that can train from scratch on small datasets. We also evaluate the performance of DHVT when training from scratch on ImageNet-1K. Our proposed DHVT-S achieves competitive 82.3% accuracy with only 24.0M parameters, which is the state-of-the-art non-hierarchical vision transformer structure as far as we know, demonstrating the effectiveness of our model on larger datasets. In summary, our main contributions are:

1. We conclude that the data efficiency on small datasets can be addressed by strengthening two inductive biases in ViTs, which are spatial relevance and diverse channel representation.

2. On spatial aspect, we adopt a hybrid model integrated with convolution, preserving fine-grained low-level features at the earlier stage and forcing the model to extract tokens feature and corresponding neighbor feature.

3. On channel aspect, we leverage the output patch tokens to re-calibrate class token channel-wise, producing better feature representation. We further introduce "head token", a novel design that helps fuse diverse feature representation encoded in different channel groups into a stronger integral representation.

## 2  Related Work

**Vision Transformers.** Convolutional Neural Networks [38, 39, 1, 40, 2, 41] dominated the computer vision fields in the past decade, with its intrinsic inductive biases designed for image recognition. The past two years witnessed the rise of Vision Transformer models in various vision tasks [42, 43, 23, 20, 44, 45]. Although there exist previous works introducing attention mechanism into CNNs [4, 46, 47], the pioneering full transformer architecture in computer vision are iGPT [48] and ViT [14]. ViT is widely adopted as the architecture paradigm for vision tasks especially image recognition. It processes the image as a token sequence and exploits relations among tokens. It uses "class token" like BERT [49] to exchange information at every layer and for final classification. It performs well when pre-trained on huge datasets. But when training from scratch on ImageNet-1K only, it underperforms ResNets, demonstrating a data-hungry problem.

**Data-efficient ViTs.** Many of the subsequent modifications on ViT strive for a more data-efficient architecture that can perform well without pre-training on larger datasets. The methods can be divided into different groups. [42, 50] use knowledge distillation strategy and stronger data-augmentation methods to enable training from scratch. [51] points out that using convolution in the patch embedding stage greatly benefits ViTs training. [52, 15, 18, 53, 54] leverage convolution for patch embedding to eliminate the discontinuity brought by non-overlapping patch embedding in vanilla ViT, and such design becomes a paradigm in subsequent works. To further introduce inductive bias into ViT, [15, 30, 34, 55, 56] integrate depth-wise convolution into feed forward network, resulting in a hybrid architecture combining the self-attention and convolution. To make ViTs more similar to standard CNNs, [16, 54, 17, 34, 33, 35, 57, 32] re-design the spatial and channel dimension of vanilla ViT, producing a series of hierarchical style vision transformer. [31, 58, 59] design another parallel convolution branch and enable the interaction with the self-attention branch, making the two branch complements each other. The above architectures introduce strong inductive bias and become data-efficient when training from scratch on ImageNet-1K. In addition, works like [60, 61, 62] investigate channel-wise representation through conducting self-attention channel-wise, while we enhance channel representation by dynamically aggregating patch token features to enhance class token channel-wise and compatibly involve channel group-wise head tokens into vanilla self-attention. Finally, works like [63, 64, 65], suggesting that the number of tokens can be variable.

**ViTs for small datasets**. There exists several works on solving the training from scratch problem on small datasets. Though the above modified vision transformers perform well when trained on ImageNet-1K, they fail to compete with standard CNNs when training on much smaller datasets like CIFAR-100. Work [66] introduces a self-supervised style training strategy and a loss function to help train ViTs on small datasets. CCT [67] adopts a convolutional tokenization module and replaces the class token with the final sequence pooling operation. SL-ViT [68] adopts shifted patch tokenization module and modifies self-attention to make it focus more locally. Though the previous works reduce the performance gap between standard CNNs ResNets[1], they fail to be sub-optimal when compared

with strong CNNs. Our proposed method leverages local constraints and enhances representation interaction, successfully bridging the performance gap on small datasets.

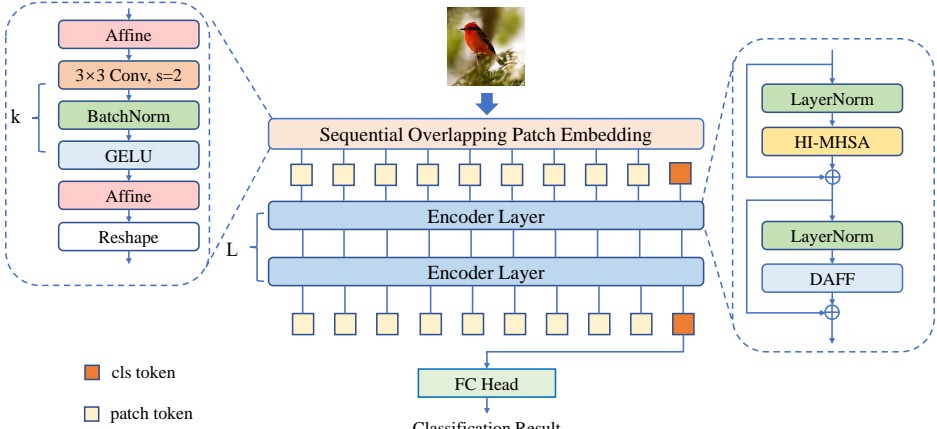

Figure 1: Overview of the proposed Dynamic Hybrid Vision Transformer (DHVT). DHVT follows a non-hierarchical structure, where each encoder layer contains two pre-norm and shortcut, a Head-Interacted Multi-Head Self-Attention (HI-MHSA) and a Dynamic Aggregation Feed Forward (DAFF).

## 3 Methods

### 3.1 Overview of DHVT

As shown in Fig. 1, the framework of our proposed DHVT is similar to vanilla ViT. We choose a non-hierarchical structure, where every encoder block shares the same parameter setting, processing the same shape of features. Under this structure, we can deal with variable length of token sequence. We keep the design of using class token to interact with all the patch tokens and for final prediction. In the patch embedding module, the input image will be split into patches first. Given the input image with resolution $H \times W$ and the target patch size $P$, the resulting length of the patch token sequence will be $N = HW/P^2$. Our modified patch embedding is called Sequential Overlapping Patch Embedding (SOPE), which contains several successive convolution layers of $3 \times 3$ convolution with stride $s = 2$, Batch Normalization and GELU [69] activation. The relation between the number of convolution layers and the patch size is $P = 2^k$. SOPE is able to eliminate the discontinuity brought by the vanilla patch embedding module, preserving important low-level features. It is able to provide position information to some extent. We also adopt two affine transformations before and after the series of convolution layers. This operation rescales and shifts the input feature, and it acts like normalization, making the training performance more stable on small datasets. The whole process of SOPE can be formulated as follows.

$$Aff(\mathbf{x}) = Diag(\boldsymbol{\alpha})\mathbf{x} + \boldsymbol{\beta} \tag{1}$$

$$G_i(\mathbf{x}) = GELU(BN(Conv(\mathbf{x}))), i = 1, \dots, k \tag{2}$$

$$SOPE(\mathbf{x}) = Reshape(Aff(G_k(\dots(G_2(G_1(Aff(\mathbf{x}))))))) \tag{3}$$

In Eq.1, $\boldsymbol{\alpha}$ and $\boldsymbol{\beta}$ are learnable parameters, and initialized as 1 and 0 respectively. After the sequence of convolution layers, the feature maps are then reshaped as patch tokens and concatenated with a class token. Then the sequence of tokens will be fed into encoder layers. After SOPE, the token sequence will pass through layers of encoder, where each encoder contains Layer Normalization [70], multi-head self-attention and feed forward network. Here we modified the MHSA as Head-Interacted Multi-Head Self-Attention (HI-MHSA) and feed forward network as Dynamic Aggregation Feed Forward (DAFF). We will introduce them in the following sections. After the final encoder layer, the output class token will be fed into the linear head for final prediction.

## 3.2 Dynamic Aggregation Feed Forward

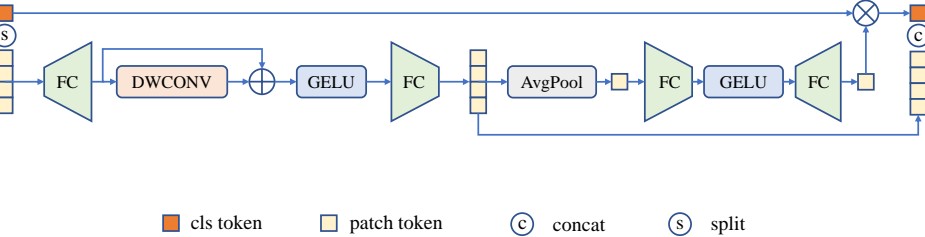

Figure 2: The structure of Dynamic Aggregation Feed Forward (DAFF).

The vanilla feed forward network (FFN) in ViT is formed by two fully-connected layers and GELU activation. All the tokens, either patch tokens or class token, will be processed by FFN. Here we integrate depth-wise convolution [36] (DWCONV) in FFN and resulting in a hybrid model. Such hybrid model is similar to standard CNNs because it can be seen as using convolution to do feature representation. With the inductive bias brought by depth-wise convolution, the model is forced to capture neighboring features, solving the problem on spatial view. It greatly reduces the performance gap when training from scratch on small datasets, and converges faster than standard CNNs. However, such a structure still performs worse than stronger CNNs. More investigations are required to solve the problem on channel aspect.

We propose two methods that make the whole model more dynamic and learn stronger feature representation under insufficient data. The first proposed module is Dynamic Aggregation Feed Forward (DAFF). We aggregate the feature of patch tokens into the class token in a channel attention way, similar to the Squeeze-Excitation operation in SENet [4], as is shown in Fig. 2. Class token is split before the projection layers. Then the patch tokens will go through a depth-wise integrated multi-layer perceptron with a shortcut inside. The output patch tokens will then be averaged into a weight vector $\mathbf{W}$. After the squeeze-excitation operation, the output weight vector will be multiplied with class token channel-wise. Then the re-calibrated class token will be concatenated with output patch tokens to restore the token sequence. We use $\mathbf{X}_c, \mathbf{X}_p$ to denote class token and patch tokens respectively. The process can be formulated as:

$$\mathbf{W} = Linear(GELU(Linear((Average(\mathbf{X}_p))))) \tag{4}$$

$$\mathbf{X}_c = \mathbf{X}_c \odot \mathbf{W} \tag{5}$$

## 3.3 Head Token

The second design to enhance feature representation is "head token", which is a brand new mechanism as far as we know. There are two reasons why we introduce head token here. First, in the original MHSA module, each attention head has not interacted with others, which means each head only focuses on itself to calculate attention. Second, channel groups in different heads are responsible for different feature representations, which is the inductive bias of ViTs. And as we pointed out above, the lack of training data can not enable models to learn strong representation. Under this circumstance, the representation in each channel group is too weak for recognition. After introducing head tokens into attention calculation, the channel group in each head are able to interact with those in other heads, and different representation can be fused into an integral representation of the object. Representation learned by insufficient data may be poor in each channel, but their combination will produce a strong enough representation. The structure of vision transformer also guarantees this mechanism because the length of input tokens is variable, except for the hierarchical structure vision transformer with window attention such as[17, 35].

The process of generating head tokens is shown in Fig. 3 (a). We denote the number of patch tokens as $N$, so the length of the input sequence is $N + 1$. According to the pre-defined number of heads $h$, each $D$-dimensional token, including class token, will be reshaped into $h$ parts. Each part contains $d$ channels, where $D = d \times h$. We average all the separated tokens in their own parts. Thus we get totally $h$ tokens and each one is $d$-dimensional. All such intermediate tokens will be projected into

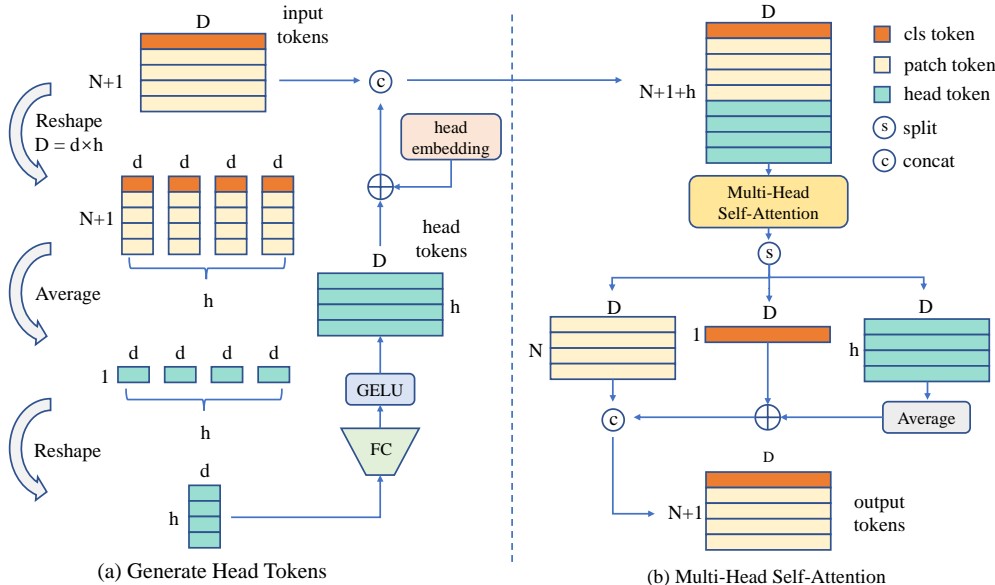

Figure 3: Pipeline of Head-Interacted Multi-Head Self-Attention (HI-MHSA).

$D$-dimension again, resulting in $h$ head tokens in total. The head tokens will be added with head embedding, which provides positional information for head tokens. Head embedding is a group of learnable parameters, just like positional embedding. Finally, they are concatenated with patch tokens and class token, forming the token sequence for standard MHSA, as Eq. 7, in which $\mathbf{X}_H$ denotes head tokens. We do not change the attention calculation in MHSA. Head tokens will also be linearly projected into query, key and value, and they will interact with all other tokens. After MHSA, the head tokens will be averaged and added to class token, just as Fig. 3 (b) shows. Head tokens can be derived as Eq. 6 shows. We use $\mathbf{E}_{head}$ to denote head embedding.

$$\mathbf{X}_H = GELU(Linear((Average(Reshape(\mathbf{X}))))) + \mathbf{E}_{head} \tag{6}$$

$$\mathbf{X} = [\mathbf{X}_c; \mathbf{X}_p; \mathbf{X}_H] = [\mathbf{X}_c; \mathbf{X}_p^1, \ldots, \mathbf{X}_p^N; \mathbf{X}_H^1, \ldots, \mathbf{X}_H^h] \tag{7}$$

## 4 Experiments

All the experiments presented in our paper are based on image classification. We do not conduct experiments on downstream tasks. We first introduce the training datasets and experimental settings in Section 4.1. The performance comparisons are shown in Section 4.2. We also show the result of the ablation study in Section 4.3. And finally, we present an example of visualization in 4.4.

### 4.1 Datasets and Experimental Settings

**Datasets.** Our main focus is training from scratch on small datasets. There are two factors to consider whether a dataset is small: the total number of training data in the dataset and the average number of training data for each class. Some datasets are small on the first factor, but large on the second. The example is CIFAR-10 [71], with 50000 training data in total for 10 classes, has an average of 5000 instances in each class. Considering this, we do not choose CIFAR-10 as our target dataset here. We choose 5 different datasets here. The main performance comparisons are on CIFAR-100 [71]. And we choose three datasets from DomainNet [37], a benchmark commonly for domain adaptation tasks. They have a large domain-shift from common medium dataset ImageNet-1K [26], making the fine-tuning experiments non-trivial, as pointed in [66]. Finally, we also choose ImageNet-1K to test the performance of our proposed model. The details of the datasets are shown in Table 1.

**Model Variants.** We propose two architecture variants. Detailed information on model variants can be seen in Supplementary Materials.

Table 1: The details of training datasets. We report the train and test size of each dataset, including the number of classes. We also show the average images per class in the training set.

| Dataset | Train size | Test size | Classes | Average images per class |
|---|---|---|---|---|
| CIFAR-100 [71] | 50000 | 10000 | 100 | 500 |
| ClipArt [37] | 33525 | 14604 | 345 | 97 |
| Sketch [37] | 48212 | 20916 | 345 | 140 |
| Painting [37] | 50416 | 21850 | 345 | 146 |
| ImageNet-1K [26] | 1281167 | 100000 | 1000 | 1281 |

- DHVT-T: 12 encoder layers, embedding dimension of 192, MLP ratios of 4, attention heads of 4 on CIFAR-100 and DomainNet, and 3 on ImageNet-1K.
- DHVT-S: 12 encoder layers, embedding dimension of 384, MLP ratios of 4, attention heads of 8 on CIFAR-100, 6 on DomainNet and ImageNet-1K.

**Implementation Details.** When training our DHVT, we keep the image size in CIFAR-100 as its original resolution $32 \times 32$, and the patch size is set to 4 or 2. For ImageNet-1K, ClipArt, Painting and Sketch, we adopt resolution $224 \times 224$, and the patch size comes to 16. All the data augmentations are the same as those in DeiT [42]. We do not tune data-augmentation hyperparameters for better performance. On all of the datasets, we train our network from random initialization with the AdamW [72] optimizer with a cosine decay learning-rate scheduler. We set the batch size of 512 and 256 for DHVT-T and DHVT-S when training on CIFAR-100, with an initial learning rate of 0.001, and a weight decay of 0.05, a warm-up epoch of 5. When on ClipArt, Sketch and Painting, we use a batch size of 256 and 128 respectively for DHVT-T and DHVT-S, with an initial learning rate of 0.001, a warm-up epoch of 20 and a weight decay of 0.05. For ImageNet-1K, we use the batch size of 512 for both models with and initial learning rate of 0.0005 and weight decay of 0.05, a warm-up epoch of 10. The training epochs on all datasets are 300, except that WRN28-10 [73] is trained for 200 epochs on CIFAR. All of the training devices are Nvidia 3090 GPUs. We use Pytorch tools and our code is modified from timm[1].

## 4.2 Performance Comparisons

Table 2: Results on DomainNet

| Method | #Params | ClipArt | Painting | Sketch |
|---|---|---|---|---|
| ResNet-50 | 24.2M | 71.90 | 64.36 | 67.45 |
| DHVT-T | 6.1M | 71.73 | 63.34 | 66.60 |
| DHVT-S | 23.8M | **73.89** | **66.08** | **68.72** |

Table 3: Results on ImageNet-1K

| Method | #Params | ImageNet-1K |
|---|---|---|
| DHVT-T | 6.2M | 76.5 |
| DHVT-S | 24.0M | 82.3 |

**Results on DomainNet.** We also conduct experiments on other small datasets. Here we choose three datasets from DomainNet as our target. We use the implementation of ResNet-50 in Pytorch official code for performance comparison. All of the data-augmentations, such as Mixup [74] and CutMix [75] and AutoAugment [76], are also adopted for training ResNet-50 from scratch on these datasets. All of the results reported are the best out of four runs. As is shown in Table 2, our model shows better results than standard ResNet-50, demonstrating its performance across different small datasets. The whole comparison of all of the DomainNet datasets with more baseline models is shown in Supplementary Materials.

**Results on ImageNet-1K** To test the train-from-scratch performance of our model on the common medium-size dataset ImageNet-1K, we also conduct experiments on it. We follow the same experimental settings as in DeiT [42]. The results are shown in Table 3. Surprisingly, our DHVT-T reaches 76.47 accuracy and our DHVT-S reaches 82.3 accuracy. As far as we know, this is the best performance under such a non-hierarchical vision transformer structure with class token. And our model outperforms many of the state-of-the-art methods with comparable parameters. This experiment shows that our model not only behaves well on small datasets but also exhibits powerful performance on larger datasets. We will show the performance comparison with other methods that train from scratch on ImageNet-1K in the Supplementary Materials.

---

[1] https://github.com/rwightman/pytorch-image-models

Table 4: Performance comparison of different methods on the CIFAR-100 dataset. All models are trained from random initialization. "⋆" denotes that we re-implement the method under the same training scheme. The other results are cited from the corresponding works.

| Type | Method | Patch Size | #Params | GFLOPs | Acc (%) |
|------|--------|------------|---------|--------|---------|
| CNN | WRN28-10 [73] | 1 | 36.5M | 5.2 | 80.75 |
| | SENet-29 [4] | 1 | 35.0M | 5.4 | 82.22 |
| | ResNeXt-29, 8×64d [40] | 1 | 34.4M | 5.4 | 82.23 |
| | SKNet-29 [41] | 1 | 27.7M | 4.2 | 82.67 |
| | DenseNet-BC (k = 40) [2] | 1 | 25.6M | 9.3 | 82.82 |
| | Res2NeXt-29, 6c×24w×6s-SE [77] | 1 | 36.9M | 5.9 | 83.44 |
| ViT | DeiT-T [42]⋆ | 4 | 5.4M | 0.4 | 67.59 |
| | DeiT-S [42]⋆ | 4 | 21.4M | 1.4 | 66.55 |
| | DeiT-T [42]⋆ | 2 | 5.4M | 1.4 | 65.86 |
| | DeiT-S [42]⋆ | 2 | 21.4M | 5.5 | 63.77 |
| | PVT-T [35] | 1 | 15.8M | 0.6 | 69.62 |
| | PVT-S [35] | 1 | 27.0M | 1.2 | 69.79 |
| | Swin-T [35] | 1 | 27.5M | 1.4 | 78.07 |
| | NesT-T [35] | 1 | 6.2M | 1.7 | 78.69 |
| | NesT-S [35] | 1 | 23.4M | 6.6 | 81.70 |
| | NesT-B [35] | 1 | 90.1M | 26.5 | 82.56 |
| Hybrid | CCT-7/3×1 [67] | 4 | 3.7M | 1.0 | 80.92 |
| | CvT-13 [18]⋆ | 4 | 19.6M | 1.1 | 79.24 |
| | CvT-13 [18]⋆ | 2 | 19.6M | 4.5 | 81.81 |
| | DHVT-T (Ours) | 4 | 6.0M | 0.4 | 80.93 |
| | DHVT-S (Ours) | 4 | 23.4M | 1.5 | 82.91 |
| | DHVT-T (Ours) | 2 | 5.8M | 1.4 | **83.54** |
| | DHVT-S (Ours) | 2 | 22.8M | 5.6 | **85.68** |

**Results on CIFAR-100.** We mainly compare the performance of our proposed model on CIFAR-100. Patch size set to 1 means taking raw pixel input. For comparison with other methods, we directly cite the results reported in the corresponding paper. The results of our model are the best out of five runs with different random seeds. As is shown in Table 4, CNN models occasionally have more parameters and conduct fewer computations, while ViT models have much fewer parameters and conduct much higher computations. Our model DHVT-T reaches 83.54 with 5.8M parameters. And DHVT-S reaches 85.68 with only 22.8M parameters. With much fewer parameters, our model achieves much higher performance against other ViT-based models and strong CNNs ResNeXt, SENet, SKNet, DenseNet and Res2Net. And compared with other ViT and Hybrid models, we exhibit a significant performance improvement under reasonable parameters and computational burdens. We not only bridge the performance gap between CNNs and ViTs but also push the state-of-the-art result to a higher level. Moreover, scaling up and smaller patch size benefit our method. Both DeiT and PVT fail to achieve higher performance when scaling up. And when the patch size gets smaller, the performance of DeiT even drops. These results are reasonable because insufficient data is hard to train a large model from scratch. And smaller patch size further intensifies the non-overlapping problem of vanilla ViT and thus decreases the performance. More experiment results such as training from scratch on 224×224 resolution can be seen in Supplementary Materials.

## 4.3 Ablation Studies

All the results in the ablation study are the average over four runs with different random seeds. The model for the ablation study is DHVT-T, with a patch size of 4 and training from scratch on CIFAR-100 with the same data augmentation as in Section 4.2. Here DHVT-T is trained with a learning rate of 0.001, a warm-up epoch of 10 and batch size of 512, and a total epoch of 300. The baseline is DeiT-T with 4 heads and the patch size is set to 4. The results are shown in the following tables.

**The importance of positional information.** We have a baseline performance of 67.59 from DeiT-T with 4 heads, training from scratch with 300 epochs. When removing absolute positional embedding, the performance drops drastically to 58.72, demonstrating the importance of position information in vision transformers. SOPE is able to provide positional information to some extent because such absolute positional information can be derived from zero padding. As is shown in Table 5, when adopting SOPE and removing absolute position embedding, the performance does not drop so drastically. But only depending on SOPE to provide position information is not enough.

Table 5: Ablation study on SOPE and DAFF

| Abs. PE | SOPE | DAFF | Acc (%) |
|---------|------|------|---------|
| ✓ | ✗ | ✗ | 67.59 (+0.00) |
| ✗ | ✗ | ✗ | 58.72 (-8.87) |
| ✓ | ✓ | ✗ | 73.68 (+6.09) |
| ✗ | ✓ | ✗ | 69.65 (+2.06) |
| ✓ | ✗ | ✓ | 79.47 (+11.88) |
| ✗ | ✗ | ✓ | 79.75 (+12.16) |
| ✓ | ✓ | ✓ | 80.17 (+12.58) |
| ✗ | ✓ | ✓ | 80.35 (+12.76) |

Table 6: Ablation study on head token

| Abs. PE | SOPE & DAFF | Head Token | Acc (%) |
|---------|-------------|------------|---------|
| ✓ | ✗ | ✗ | 67.59 (+0.00) |
| ✓ | ✗ | ✓ | 69.10 (+1.51) |
| ✗ | ✓ | ✓ | 80.85 (+13.26) |

**The role of DAFF.** When adopting DAFF, the performance gain increases greatly to 79.47, because DAFF solves the problem on both spatial and channel aspects, introducing strong local constraints and re-calibrating channel feature representation. It is sensible to see that removing absolute position embedding can increase performance. The positional information has been encoded into tokens through the depth-wise convolution in DAFF, and the absolute position embedding will break translation invariance. When both SOPE and DAFF are adopted, the positional information will be encoded comprehensively, and SOPE will also help address the non-overlapping problem here, preserving fine-grained low-level features in the early stage.

**The role of head tokens.** From Table 6, we can also see the stable performance gain brought by head tokens across different model structures. When introducing head tokens into DeiT-T, the performance gets a +1.51 gain, demonstrating its effectiveness. As we said before, head tokens guarantee the interaction among different channel groups, better fusing the diverse representation. The resulting integral representation is now strong enough for classification. When adopting all three modifications, we get a +13.26 accuracy gain, successfully bridging the performance gap with CNNs.

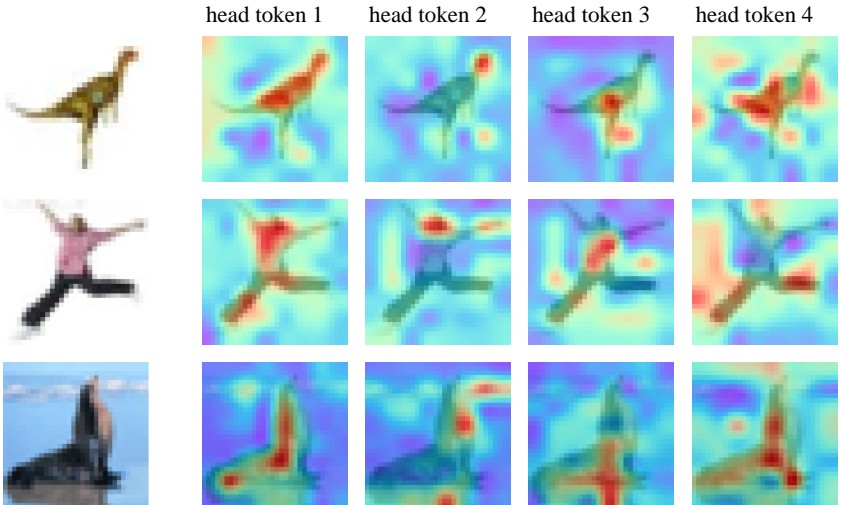

Figure 4: Visualization of the attention map of head tokens to patch tokens on low layer

### 4.4 Visualization

We visualize the attention maps of head tokens to patch tokens in Fig. 4. Each row represents one image. The results are samples in the second encoder layer. We can see that different head token activates on different patch tokens, exhibiting their diverse representations. On such low layers, low-level fine-grained features are able to be captured in our model. More visualization results are shown in the Supplementary Materials.

## 5  Limitation

Though we achieve a much higher performance than existing methods, such performance gain comes at the expense of computation. The performance when patch size set to 2 boosts higher than using patch size of 4. But the computation expense rises quadratically. In practical usage, we suggest choose a good patch size for better trade-off between performance and computation.

## 6  Conclusion

In this paper, we present an alternative vision transformer architecture DHVT, which can train from scratch on small datasets and reach state-of-the-art performance on a series of datasets. The weak inductive biases of spatial relevance and diverse channel representation brought by insufficient training data are strengthened in our model. The highlighted head token design is able to transfer to variants of ViT model to enable better feature representation.

## 7  Acknowledgements

This work is supported by the National Nature Science Foundation of China (62121002, 62022076, 62232006, U1936210, 62272436), the Youth Innovation Promotion Association Chinese Academy of Sciences (Y2021122), Anhui Provincial Natural Science Foundation (2208085QF190), the China Postdoctoral Science Foundation 2021M703081.

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
