# Bridging the Gap Between Vision Transformers and Convolutional Neural Networks on Small Datasets–Supplementary Materials

**Zhiying Lu, Hongtao Xie\*, Chuanbin Liu\*, Yongdong Zhang**
University of Science and Technology of China, Hefei, China
arieseirack@mail.ustc.edu.cn, {htxie,liucb92,zhyd73}@ustc.edu.cn

## 1 Pseudo Code of Dynamic Hybrid Vision Transformer

### 1.1 Pseudo Code of the Dynamic Aggregation Feed Forward (DAFF)

```python
class DAFF(nn.Module):
    def __init__(self, in_dim, hid_dim, out_dim, kernel_size=3):
        self.conv1 = nn.Conv2d(in_dim, hid_dim, kernel_size=1,
                               stride=1, padding=0)
        self.conv2 = nn.Conv2d(
            hid_dim, hid_dim, kernel_size=3, stride=1,
            padding=(kernel_size-1)//2, groups=hid_dim)
        self.conv3 = nn.Conv2d(hid_dim, out_dim, kernel_size=1,
                               stride=1, padding=0)
        self.act = nn.GELU()
        self.squeeze = nn.AdaptiveAvgPool2d((1, 1))
        self.compress = nn.Linear(in_dim, in_dim//4)
        self.excitation = nn.Linear(in_dim//4, in_dim)
        self.bn1 = nn.BatchNorm2d(hid_dim)
        self.bn2 = nn.BatchNorm2d(hid_dim)
        self.bn3 = nn.BatchNorm2d(out_dim)

    def forward(self, x):
        B, N, C = x.size()
        cls_token, tokens = torch.split(x, [1, N-1], dim=1)
        x = tokens.reshape(B, int(math.sqrt(N-1)),
                int(math.sqrt(N-1)), C).permute(0, 3, 1, 2)

        x = self.act(self.bn1(self.conv1(x)))
        x = x + self.act(self.bn2(self.conv2(x)))
        x = self.bn3(self.conv3(x))

        weight = self.squeeze(x).flatten(1).reshape(B, 1, C)
        weight = self.excitation(self.act(self.compress(weight)))
        cls_token = cls_token * weight
        tokens = x.flatten(2).permute(0, 2, 1)
        out = torch.cat((cls_token, tokens), dim=1)
        return out
```

[1]

---

\*Corresponding author
[1]Code is modified from https://github.com/coeusguo/ceit

36th Conference on Neural Information Processing Systems (NeurIPS 2022).

## 1.2 Pseudo Code of the Sequential Overlapping Patch Embedding (SOPE)

```python
def conv3x3(in_dim, out_dim):
    return torch.nn.Sequential(
        nn.Conv2d(in_dim,out_dim,kernel_size=3,stride=2,padding=1),
        nn.BatchNorm2d(out_dim)
    )

class Affine(nn.Module):
    def __init__(self, dim):
        self.alpha = nn.Parameter(torch.ones([1, dim, 1, 1]))
        self.beta = nn.Parameter(torch.zeros([1, dim, 1, 1]))
    def forward(self, x):
        x = x * self.alpha + self.beta
        return x

class SOPE(nn.Module):
    def __init__(self, patch_size, embed_dim):
        self.pre_affine = Affine(3)
        self.post_affine = Affine(embed_dim)
        if patch_size[0] == 16:
            self.proj = torch.nn.Sequential(
                conv3x3(3, embed_dim//8, 2),
                nn.GELU(),
                conv3x3(embed_dim//8, embed_dim//4, 2),
                nn.GELU(),
                conv3x3(embed_dim//4, embed_dim//2, 2),
                nn.GELU(),
                conv3x3(embed_dim//2, embed_dim, 2),
                )
        elif patch_size[0] == 4:
            self.proj = torch.nn.Sequential(
                conv3x3(3, embed_dim//2, 2),
                nn.GELU(),
                conv3x3(embed_dim//2, embed_dim, 2),
                )
        elif patch_size[0] == 2:
            self.proj = torch.nn.Sequential(
                conv3x3(3, embed_dim, 2),
                nn.GELU(),
                )
    def forward(self, x):
        B, C, H, W = x.shape
        x = self.pre_affine(x)
        x = self.proj(x)
        x = self.post_affine(x)
        Hp, Wp = x.shape[2], x.shape[3]
        x = x.flatten(2).transpose(1, 2)
        return x
```

2

---

[2]Code is modified from https://github.com/facebookresearch/xcit

## 1.3 Pseudo Code of the Head-Interacted Multi-Head Self-Attention (HI-MHSA)

```python
class Attention(nn.Module):
    def __init__(self, dim, num_heads=8):
        super().__init__()
        self.num_heads = num_heads
        head_dim = dim // num_heads
        self.scale = head_dim ** -0.5
        self.qkv = nn.Linear(dim, dim * 3, bias=True)
        self.proj = nn.Linear(dim, dim)
        self.act = nn.GELU()
        self.ht_proj = nn.Linear(head_dim, dim, bias=True)
        self.ht_norm = nn.LayerNorm(head_dim)
        self.pos_embed = nn.Parameter(
                    torch.zeros(1, self.num_heads, dim))

    def forward(self, x):
        B, N, C = x.shape

        # head token
        head_pos = self.pos_embed.expand(x.shape[0], -1, -1)
        ht = x.reshape(B, -1, self.num_heads, C//self.num_heads).
                                            permute(0, 2, 1, 3)
        ht = ht.mean(dim=2)
        ht = self.ht_proj(ht)
                    .reshape(B, -1, self.num_heads, C//self.num_heads)
        ht = self.act(self.ht_norm(ht)).flatten(2)
        ht = ht + head_pos
        x = torch.cat([x, ht], dim=1)

        # common MHSA
        qkv = self.qkv(x).reshape(B, N+self.num_heads, 3,
                    self.num_heads, C//self.num_heads)
                    .permute(2, 0, 3, 1, 4)
        q, k, v = qkv[0], qkv[1], qkv[2]
        attn = (q @ k.transpose(-2, -1)) * self.scale
        attn = attn.softmax(dim=-1)
        attn = self.attn_drop(attn)
        x = (attn @ v).transpose(1, 2).reshape(B, N+self.num_heads, C)
        x = self.proj(x)

        # split, average and add
        cls, patch, ht = torch.split(x, [1,N-1,self.num_heads], dim=1)
        cls = cls + torch.mean(ht, dim=1, keepdim=True)
        x = torch.cat([cls, patch], dim=1)

        return x
```

## 2 CIFAR-100 Dataset

### 2.1 Fine-tuning on CIFAR-100

We analyze fine-tuning results in this section. All the models are pre-trained on ImageNet-1K [1] only and then fine-tuned on CIFAR-100 [2] datasets. Results are shown in Table 1. We cite the reported results from corresponding papers. When fine-tuning our DHVT, we use AdamW optimizer with cosine learning rate scheduler and 2 warm-up epochs, a batch size of 256, an initial learning rate of 0.0005, weight decay of 1e-8, and fine-tuning epochs of 100. We fine-tune our model on the image size of 224×224 and we use a patch size of 16, head numbers of 3 and 6 for DHVT-T and DHVT-S respectively, the same as the pre-trained model on ImageNet-1K.

Table 1: Pretrained on ImageNet-1K and then fine-tuned on the CIFAR-100 (top-1 accuracy, 100 fine-tuning epochs). "FT Epochs" denotes fine-tuning epochs, and "Img Size" denotes the input image size in fine-tuning.

| Method | GFLOPs | ImageNet-1K | FT Epochs | Img Size | CIFAR-100 Acc (%) |
|---|---|---|---|---|---|
| ResNet-50 [3] | 3.8 | - | 100 | 224 | 85.44 |
| ViT-B/16 [4] | 18.7 | 77.9 | 10000 | 384 | 87.13 |
| ViT-L/16 [4] | 65.8 | 76.5 | 10000 | 384 | 86.35 |
| T2T-ViT-14 [3] | 5.2 | 81.5 | 100 | 224 | 87.33 |
| Swin-T [3] | 4.5 | 81.3 | 100 | 224 | 88.22 |
| DeiT-B [5] | 17.3 | 81.8 | 7200 | 224 | 90.8 |
| DeiT-B ↑384 [5] | 52.8 | 83.1 | 7200 | 384 | 90.8 |
| CeiT-T [6] | 1.2 | 76.4 | 100 | 224 | 88.4 |
| CeiT-T ↑384 [6] | 3.6 | 78.8 | 100 | 384 | 88.0 |
| CeiT-S [6] | 4.5 | 82.0 | 100 | 224 | 90.8 |
| CeiT-S ↑384 [6] | 12.9 | 83.3 | 100 | 384 | 90.8 |
| DHVT-T (Ours) | 1.2 | 76.5 | 100 | 224 | 86.73 |
| DHVT-S (Ours) | 4.7 | 82.3 | 100 | 224 | 88.87 |

From Table 1, we can see that our model has competitive transferring performance with Swin Transformer, T2T-ViT. We fail in competing with DeiT [5] maybe because we only fine-tuned our model for 100 epochs. The longer epochs experiments are left for the future. And we also fail to compete with CeiT [6] under comparable computational complexity. We consider that maybe our model introduced too many inductive biases so that the fine-tuning performance is constrained also. However, the target of our method is mainly on train-from-scratch on small datasets. Thus the fine-tuning results are not so important in our consideration. And we can also see that we achieve 85.68 accuracy when training from scratch on CIFAR-100 only, as we reported in the main part of this paper. Such a result even outperforms ResNet-50 pre-trained on ImageNet-1k, which only reaches 85.44 accuracy when fine-tuning. DHVT can beat the pre-trained and fine-tuned ResNet-50 without any pre-training, suggesting the significant performance of our model.

### 2.2 More Ablation Studies

In this part, we present more ablation studies on the minor operation or module variant in our method. The experimental setup is the same as the main paper. We present ablation study results on DAFF, SOPE, the number of attention heads, and the choice between Batch Normalization and Layer Normalization.

#### 2.2.1 Minor Operations in DAFF

We first present the ablation study on the minor operation in DAFF to show why we finally adopt such a structure. The influences of minor operation on the Feed Forward Network (FFN) in the original ViT are shown in Table 2. (1) "Split CLS" means that the class token is split away from other patch tokens and it will pass through FFN without any computation. And we can see a +1.17 improvement in accuracy. It may imply that the class token contains the global information of the image and that the feature carried by the class token is quite different from other patch tokens. Thus

if both class token and patch tokens are projected by the same FFN, it would be hard to train the model. (2) "Agg on CLS" means using Squeeze-Excitation [7] operation to dynamically aggregate information from patch tokens and re-calibrate class token channel-wise. And we can see a further improvement in the accuracy. This is a reasonable way to re-calibrate the class token because it carries global information. We then use the global average pooling to gather information from patch tokens and refine the feature to enhance class token. (3) "AvgPool" means using Average Pooling instead of Depth-wise Convolution (DWCONV)[8] to process the patch tokens. And we can see a non-trivial improvement. Average Pooling can be seen as a deteriorated version of DWCONV. The improvement brought by AvgPool implies that attending to and aggregating neighboring features is indeed helpful and necessary.

Table 2: Ablation study on the Minor Operation on FFN of ViT

| Variants | Acc (%) |
|---|---|
| ViT | 67.59 (+0.00) |
| ViT + Split CLS | 68.76 (+1.17) |
| ViT + Split CLS + Agg on CLS | 69.34 (+1.75) |
| ViT + Split CLS + AvgPool | 70.48 (+2.89) |

We then conduct an ablation study on the influence brought by minor operations upon the full version of DHVT. The results are shown in Table 3. The baseline DHVT-T with a patch size of 4 achieves 80.85 accuracy on CIFAR-100. (1) When the dynamic aggregation operation re-calibrates all the tokens, rather than only the class token, the performance will drop drastically. (2) When removing the shortcut alongside the DWCONV, the performance also drops. The shortcut retains the original feature, and thus the feature holds global and local information simultaneously.

Table 3: Ablation study on the Minor Operation in DAFF

| Variants | Acc (%) |
|---|---|
| DHVT | 80.85 (+0.00) |
| DHVT (Agg on all token) | 78.34 (-2.51) |
| DHVT (w/o shortcut) | 80.14 (-0.71) |

### 2.2.2 Affine Operation in SOPE

We introduce two affine transformations before and after the sequential convolution layers. The pre-affine is done by transforming the original input and the post-affine is to adjust the feature after the convolution sequence. This method renders stable training results on the CIFAR-100 dataset. If we remove such an operation, the average performance will drop from 80.90 to 80.72. On the Vanilla ViT with SOPE, the performance is 73.68, and if removing the two affine transformations, the accuracy will drop to 73.42.

### 2.2.3 The Number of Attention Head

It is well-known that DeiT-Tiny and DeiT-Small have 3 and 6 heads respectively. However, this parameter is chosen by the experiment on ImageNet. Similarly, we set the number of heads as 4 and 8 for our DHVT-T and DHVT-S based on the experiment on CIFAR-100. The results can be seen in Table 4. To be compatible with scalability, we adopt 4 heads in DHVT-T and 8 heads in DHVT-S. We hypothesize this is because each attribute of the object in CIFAR does not need too many channels for representation. So we keep the number of channels in each head less than usual.

### 2.2.4 The choice of Batch Normalization and Layer Normalization

Here in our work, BatchNorm is adopted at two positions: SOPE and DAFF. We use (A-B) to denote normalization choice, where A is the normalization operation in SOPE and B is in DAFF. The experiment is conducted on DHVT-T. We evaluate the influence from a batch size of 128, 256 and 512. From Table 5, we can see that using BN is indeed sensitive to the batch size, while its performance is always superior to LN. From our point of view, the vanilla ViT adopts LN before Multi-Head Self-Attention (MHSA) and Multi-layer Perceptron (MLP), aiming at regularizing each token on channel dimension. This is important because MHSA uses dot-product operation, and LN

Table 4: The influence of number of attention head on CIFAR-100 dataset.

| Method | #Head | Acc (%) |
|--------|-------|---------|
| ViT-T | 3 | 66.50 |
| | 4 | **67.59** |
| DHVT-T | 3 | 80.92 |
| | 4 | **80.98** |
| DHVT-S | 4 | 82.27 |
| | 6 | 82.59 |
| | 8 | **82.82** |

helps control the value of the query and key, avoiding extreme values. In our work, LN is also adopted at the same place, before MHSA and MLP. Further, we use convolution operation, and its output should be regularized in terms of spatial dimension. When we replace BN with LN, the feature will be reshaped into a sequence style, which may ignore the spatial relations. So it is more suitable to use BN along with convolution for higher recognition accuracy.

Table 5: The influence of the choice between Batch Normalization (BN) and Layer Normalization (LN) on CIFAR-100. "Var" and "BS" denote the variant and the batch size respectively.

| Acc (%) \ BS \ Var | 128 | 256 | 512 |
|--------------------|-----|-----|-----|
| BN-BN | 79.69 | 80.31 | **80.98** |
| BN-LN | 78.69 | 79.55 | 79.25 |
| LN-BN | 79.24 | 80.02 | 80.26 |
| LN-LN | 78.46 | 79.09 | 78.90 |

## 2.3 Training Efficiency on CIFAR-100

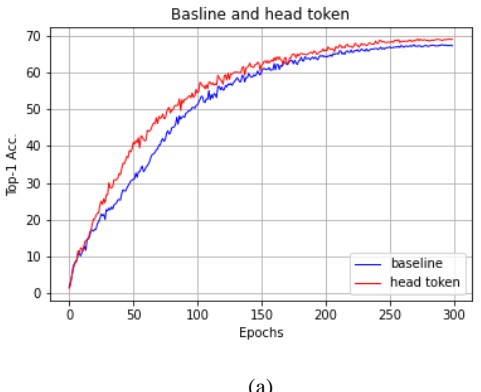

(a)

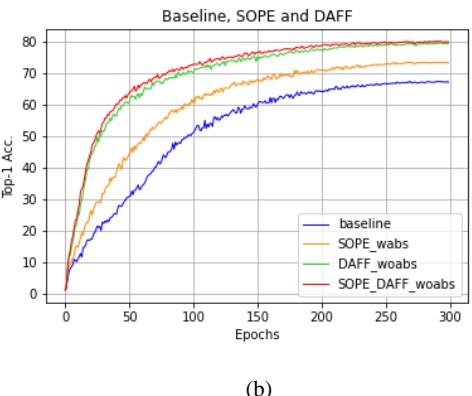

(b)

Figure 1: (a) Training Efficiency of applying head token only. (b) Training efficiency of applying SOPE and DAFF and with or without absolute positional embedding.

In this section, we show the training efficiency of our method when applying each module. All the experimental settings are the same as in the ablation study section. The baseline module is DeiT-Tiny with 4 heads. Here we denote "without absolute positional embedding" as "woabs" and "wabs" denotes the opposite. From Fig. 1 (a), we can see that introducing head token into the baseline can facilitate the overall training process and reaches higher accuracy, proving the effectiveness of our novel head token design. And from Fig. 1 (b), we can see that both SOPE and DAFF can improve the whole training, and their combination has a positive influence on the model.

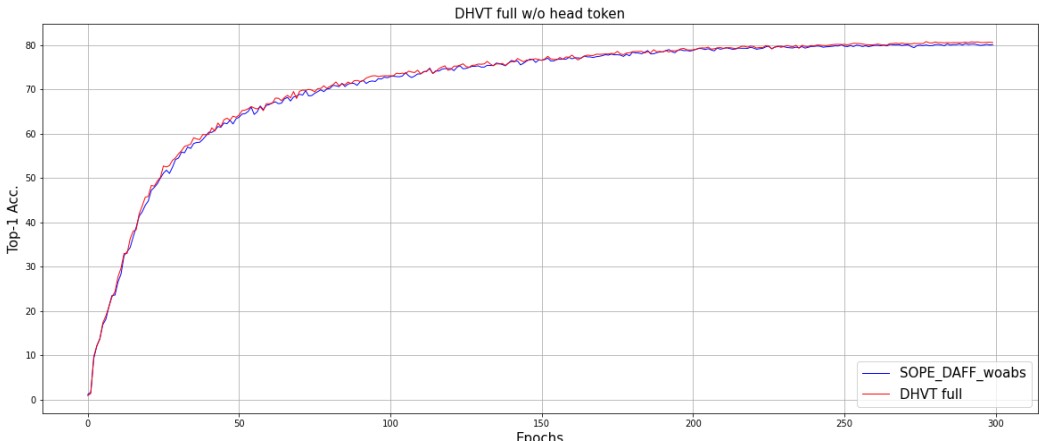

Figure 2: Training efficiency of head token when both SOPE and DAFF are applied and absolute positional embedding is removed.

In addition, in Fig. 2, we use "DHVT full" to represent the full version of our model, including SOPE, DAFF and head tokens, while removing absolute positional embedding. In such a circumstance, head token is still able to give rise to the performance during most of the training epochs, and the final result is also a little higher than the one without head token.

## 2.4 Visualization on CIFAR-100

To further understand the feature interaction style in our proposed model, we provide more visualization results in this section. First, we visualize the averaged attention map of all the tokens, including class token, patch tokens and head tokens on the 2nd, 5th, 8th and 11th encoder layers. We provide three example input images in total. Second, we visualize the attention of head tokens to patch tokens in their corresponding head on the 2nd, 5th, 8th and 11th encoder layers. The model we visualize here is DHVT-T training from scratch on the CIFAR-100 dataset, which contains 4 attention heads in each layer and thus the corresponding number of head tokens is 4. And patch size is set to 4 here.

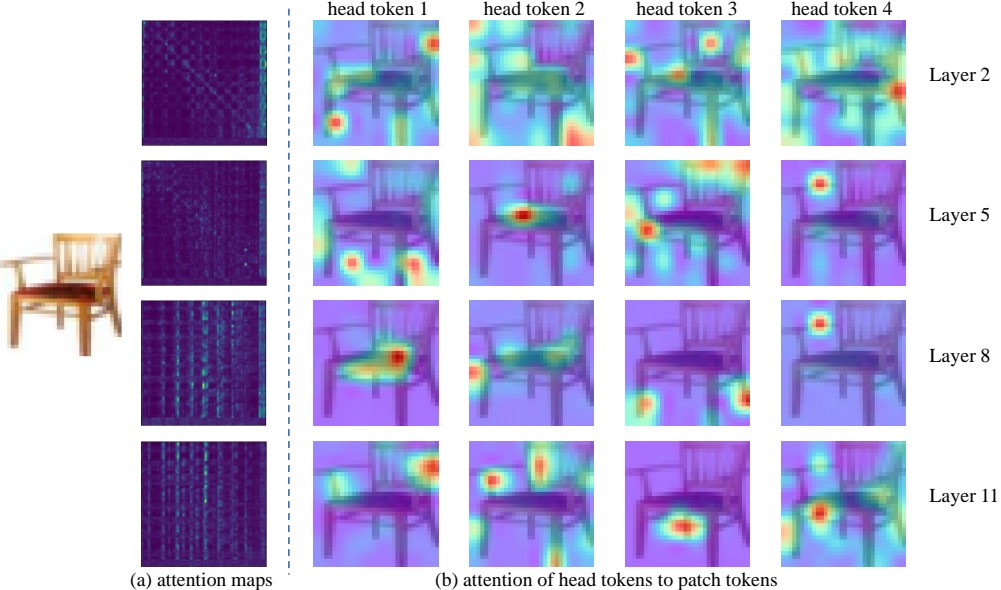

Figure 3: Visualization on CIFAR-100. (a) Averaged attention maps. (b) Attention of head tokens to patch tokens in the corresponding heads.

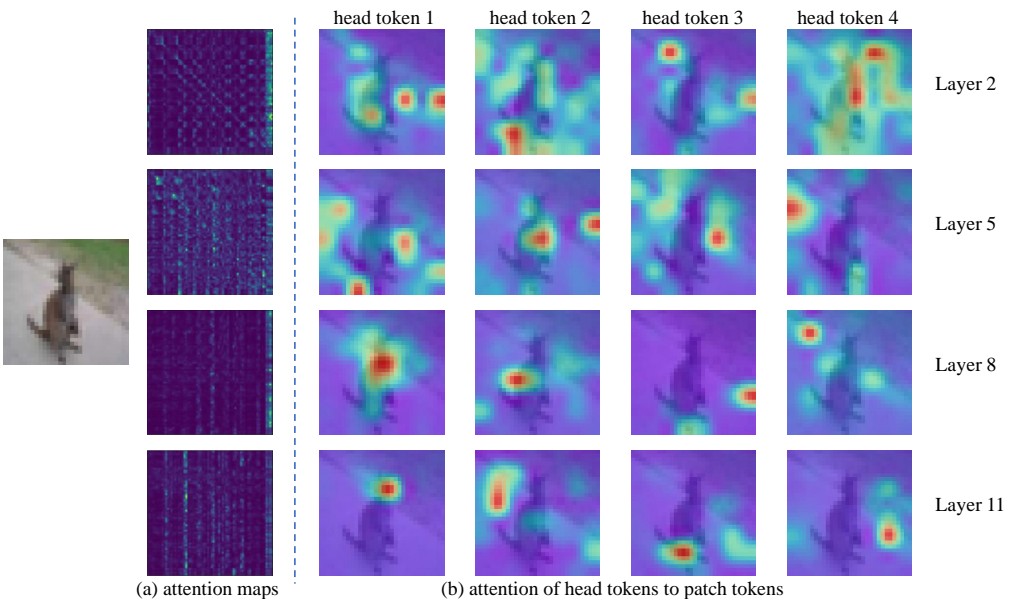

Figure 4: Visualization on CIFAR-100. (a) Averaged attention maps. (b) Attention of head tokens to patch tokens in the corresponding heads.

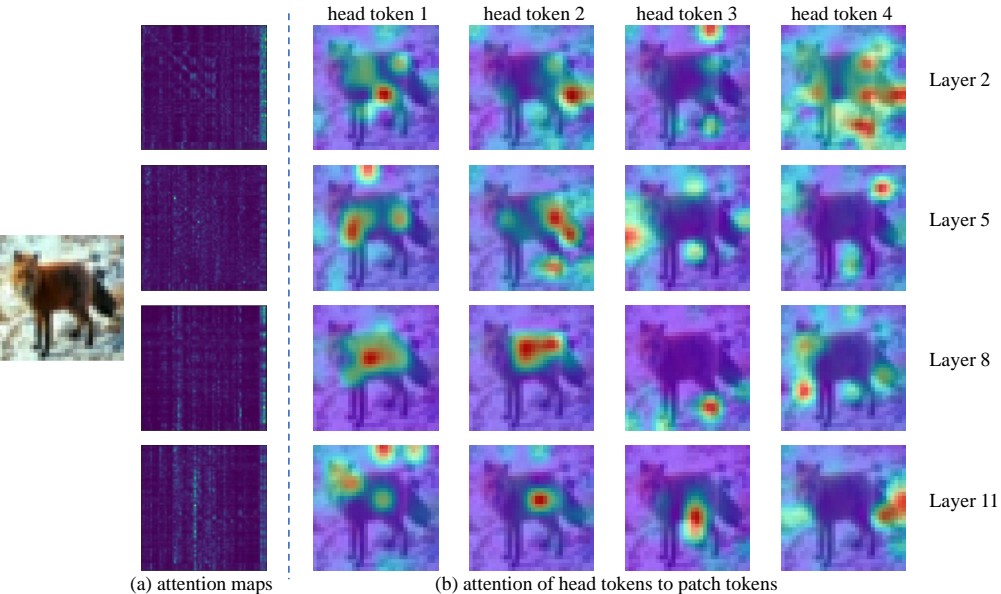

Figure 5: Visualization on CIFAR-100. (a) Averaged attention maps. (b) Attention of head tokens to patch tokens in the corresponding heads.

Note that head tokens are concatenated behind patch tokens, so the right-hand side of attention maps represents the attention from all the tokens to head tokens. From the above results in Figure 3,4,5, we can summarize two attributes that head tokens brought to the model. First, in the lower layers, such as the 2nd layer, the model tends to attend to neighboring features and interacts with head tokens. Going deeper, such as in the 5th layer, attention is scattered around all tokens and head tokens do not receive much attention here. In higher layers, like in the 8th layer, attention focus on some of the patch tokens and now head tokens receive more attention than in the 5th layer. Finally, in the layers near the output layer, such as the 11th layer, patch tokens do not focus too much on head tokens, and all the tokens converge their attention to the most prominent patch tokens.

Second, each head token represents a different representation as we visualized above. When head tokens participate in attention calculation, they help the interaction of different representations, fusing poor representation encoded in different channel groups into a strong integral representation. The results are similar on ImageNet-1K and we provide a discussion later.

## 3 DomainNet Dataset

### 3.1 Examples of DomainNet

In this part, we visualize some example images in the DomainNet [9] datasets as in Fig. 6. These datasets have a domain shift from traditional natural image datasets like ImageNet-1K and CIFAR. Also because of the scarce training data, models are hard to train from scratch on such datasets. However, our proposed DHVT can address the issue with satisfactory results in both train-from-scratch and pretrain-finetune scenario. Under a comparable amount of computational complexity, our models exhibit non-trivial performance gain compared with baseline models on all of the six datasets.

Table 6: The statistics of training datasets. We report the train and test size of each DomainNet dataset, including the number of classes. We also show the average images per class in the training set.

| Dataset | Train size | Test size | Classes | Average images per class |
|---|---|---|---|---|
| ClipArt [9] | 33525 | 14604 | 345 | 97 |
| Sketch [9] | 48212 | 20916 | 345 | 140 |
| Painting [9] | 50416 | 21850 | 345 | 146 |
| Infograph [9] | 36023 | 15582 | 345 | 104 |
| Real [9] | 120906 | 52041 | 345 | 350 |
| Quickdraw [9] | 120750 | 51750 | 345 | 350 |

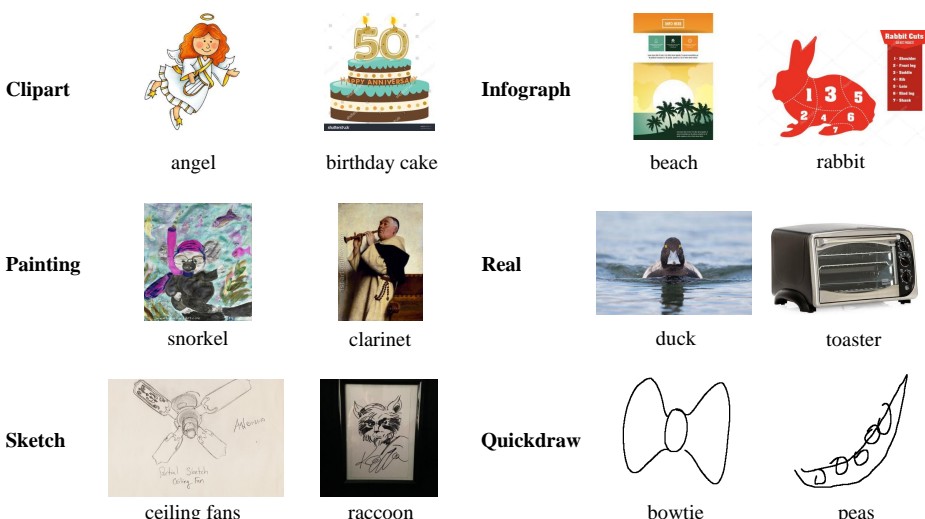

Figure 6: Visualization of examples in each DomainNet dataset.

## 3.2 Comprehensive Train-from-scratch Results on DomainNet

In this section, we present the whole performance comparison of train-from-scratch over all the DomainNet datasets between our model and baseline models. The training scheme is shown in the main paper in Section 4.1. From Table 7, our method demonstrates a consistent performance gap over baseline models.

Table 7: Comprehensive Results on DomainNet. All the models are trained from scratch for 300 epochs under the same training schedule. The training resolution is 224×224. "C", "P", "S", "I", "R", "Q" denotes the accuracy of ClipArt, Painting, Sketch, Infograph, Real and Quickdraw datasets respectively. And we adopt ResNeXt-50, 32x4d variant.

| Method | #Params | GFLOPs | C | P | S | I | R | Q |
|---|---|---|---|---|---|---|---|---|
| ResNet-50 [10] | 24.2M | 3.8 | 71.90 | 64.36 | 67.45 | 32.40 | 81.51 | 74.19 |
| ResNeXt-50 [11] | 23.7M | 4.3 | 72.93 | 64.37 | 68.52 | 34.85 | 82.15 | 73.73 |
| CvT-13 [12] | 19.7M | 4.5 | 69.77 | 61.57 | 66.16 | 30.07 | 81.48 | 72.49 |
| DHVT-T | 6.1M | 1.2 | 71.73 | 63.34 | 66.60 | 32.60 | 81.31 | **74.41** |
| DHVT-S | 23.8M | 4.7 | **73.89** | **66.08** | **68.72** | **35.11** | **83.64** | 74.38 |

## 3.3 Fine-tuning on DomainNet

We analyze the fine-tuning results on DomainNet datasets in this section. All the models are pre-trained on ImageNet-1K only and then fine-tuned on Clipart, Painting, and Sketch. Results are shown in Table 8. We cite the reported results from corresponding papers. Note that the fine-tuning epochs in baseline models are 100, the same as we use. When fine-tuning our DHVT, we use AdamW optimizer with cosine learning rate scheduler and 2 warm-up epochs, a batch size of 256, an initial learning rate of 0.0005, weight decay of 1e-8, and fine-tuning epochs of 100. We fine-tune our model on the image size of 224×224 and we use a patch size of 16, head numbers of 3 and 6 for DHVT-T and DHVT-S respectively, the same as the pre-trained model on ImageNet-1K.

Table 8: Pretrained on ImageNet-1K and then fine-tuned on the DomainNet (top-1 accuracy (%), 100 fine-tuning epochs). The ImageNet-1K column shows the accuracy of pretrained model on ImageNet-1K.

| Method | GFLOPs | ImageNet-1K | Clipart | Painting | Sketch |
|---|---|---|---|---|---|
| ResNet-50 [3] | 3.8 | - | 75.22 | 66.58 | 67.77 |
| T2T-ViT-14 [3] | 5.2 | 81.5 | 74.59 | 72.29 | 72.18 |
| Swin-T [3] | 4.5 | 81.3 | 73.51 | 72.99 | 72.37 |
| DHVT-T (Ours) | 1.2 | 76.5 | 77.88 | 72.05 | 70.79 |
| DHVT-S (Ours) | 4.7 | 82.3 | **80.06** | **74.18** | **73.32** |

From Table 8, we can see that our models show better performance than baseline methods ResNet-50, Swin Transformer, and T2T-ViT. Especially on Clipart, our DHVT-S reaches more than 80 accuracy, showing a significantly better performance than baseline methods. Our tiny model achieves comparable and even better accuracy than T2T-ViT and Swin Transformer with much lower computational complexity on Clipart and Painting. From the main part of this paper, the performance of training from scratch of DHVT-S is 68.72, as shown in the main part of this paper, which outperforms the fine-tuning result of ResNet-50, exhibiting the train-from-scratch capacity of our method.

# 4 ImageNet-1K Dataset

## 4.1 Comparison on ImageNet-1K

We conduct experiments on ImageNet-1K dataset to test the performance of our proposed DHVT on the common medium dataset. From the above Table 9, we can see that with fewer parameters and comparable computational complexity, our DHVT achieves state-of-the-art results compared to

Table 9: Performance comparison of different methods on ImageNet-1K. All models are trained from random initialization.

| Method | #Params | Image Size | GFLOPs | Top-1 Acc (%) |
|---|---|---|---|---|
| RegNetY-800MF [13] | 6.3M | 224 | 0.8 | 76.3 |
| RegNetY-4.0GF [13] | 20.6M | 224 | 4.0 | 79.4 |
| ConvNeXt-T [14] | 29M | 224 | 4.5 | 82.1 |
| T2T-ViT-7 [15] | 4.3M | 224 | 1.2 | 71.7 |
| DeiT-T [5] | 5.7M | 224 | 1.1 | 72.2 |
| PiT-Ti [16] | 4.9M | 224 | 0.7 | 72.9 |
| ConViT-Ti [17] | 5.7M | 224 | 1.4 | 73.1 |
| CrossViT-Ti [18] | 6.9M | 224 | 1.6 | 73.4 |
| TNT-T [19] | 6.2M | 224 | 1.4 | 73.6 |
| LocalViT-T [20] | 5.9M | 224 | 1.3 | 74.8 |
| ViTAE-T [21] | 4.8M | 224 | 1.5 | 75.3 |
| CeiT-T [6] | 6.4M | 224 | 1.2 | 76.4 |
| DHVT-T (Ours) | 6.2M | 224 | 1.2 | **76.5** |
| DeiT-S [5] | 22.1M | 224 | 4.3 | 79.8 |
| PVT-S [22] | 24.5M | 224 | 3.8 | 79.8 |
| PiT-S [16] | 23.5M | 224 | 2.9 | 80.9 |
| CrossViT-S [18] | 26.7M | 224 | 5.6 | 81.0 |
| PVT-Medium [22] | 44.2M | 224 | 6.7 | 81.2 |
| Conformer-Ti [23] | 23.5M | 224 | 5.2 | 81.3 |
| Swin-T [24] | 29.0M | 224 | 4.5 | 81.3 |
| ConViT-S [17] | 27.8M | 224 | 5.4 | 81.3 |
| TNT-S [19] | 23.8M | 224 | 5.2 | 81.3 |
| T2T-ViT-14 [15] | 21.5M | 224 | 5.2 | 81.5 |
| NesT-T [25] | 17.0M | 224 | 5.8 | 81.5 |
| CvT-13 [12] | 20.0M | 224 | 4.5 | 81.6 |
| Twins-SVT-S [26] | 24.0M | 224 | 2.8 | 81.7 |
| CaiT-XS24 [27] | 26.6M | 224 | 5.4 | 81.8 |
| CoaT-Lite Small [28] | 20.0M | 224 | 4.0 | 81.9 |
| CeiT-S [6] | 24.2M | 224 | 4.5 | 82.0 |
| ViL-S [29] | 24.6M | 224 | 4.9 | 82.0 |
| PVTv2-B2 [30] | 25.4M | 224 | 4.0 | 82.0 |
| ViTAE-S [21] | 23.6M | 224 | 5.6 | 82.0 |
| LG-T [31] | 32.6M | 224 | 4.8 | 82.1 |
| Focal-T [32] | 29.1M | 224 | 4.9 | 82.2 |
| DHVT-S (Ours) | 24.1M | 224 | 4.7 | **82.3** |

recent CNNs and ViTs. Both of our models are trained from scratch on ImageNet-1K datasets, with an image size of 224×224, patch size of 16, optimizer of AdamW, and base learning rate of 0.0005 following cosine learning rate decay, weight decay of 0.05, a warm-up epoch of 10, batch size of 512. All the data-augmentations and regularizations methods follow Deit [5], including random cropping, random flipping, label-smoothing [33], Mixup [34], CutMix [35] and random erasing [36].

Our DHVT-T reaches 76.5 accuracy with only 6.2M parameters, while DHVT-S achieves 82.3 accuracy with only 24.1M parameters. Our model not only outperforms the best non-hierarchical vision transformer CeiT [6] but also shows competitive performance to most of the hierarchical vision transformers like Swin Transformer [24] and hybrid architecture like ViTAE-S [21]. We also show better performance than recent strong CNNs RegNet [13] and ConvNeXt [14]. We achieve such results with much fewer parameters than existing methods, while our computational complexity is also higher than theirs. This is a kind of mixed blessing. On one hand, our method can be seen as using fewer parameters to conduct comprehensive and sufficient computation. On the other hand, such an amount of computation is a huge burden for both training and testing. We hope to reduce the computational burden in future research while maintaining the same performance.

## 4.2  Visualization on ImageNet-1K

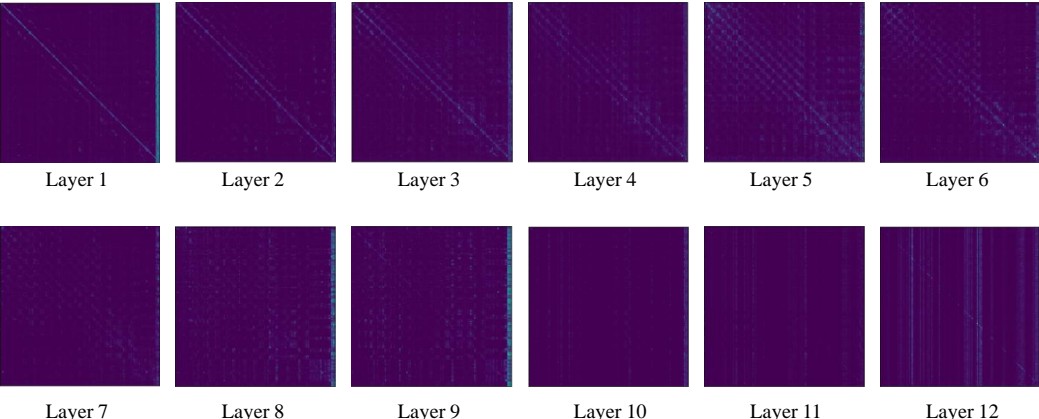

Figure 7: Averaged attention maps from DHVT-S training from scratch on ImageNet-1K.

We further visualize the attention maps of our proposed model training from scratch on ImageNet-1K only. Here the model is DHVT-S with 6 attention heads and a patch size of 16. Note that head tokens are concatenated behind patch tokens, so the right-hand side of attention maps represents the attention from all the tokens to head tokens. With the introduction of head tokens, we can understand the feature extraction and representation mechanism in our model.

In the input encoder layer, i.e. the 1st layer, all the tokens focus on themselves and the head tokens. And in the early stage, i.e. from the 2nd to 6th layers, all the tokens focus more on themselves and do not attend too much to head tokens. Further, in the middle stage, i.e. from the 7th to 9th layers, head tokens draw more attention from other tokens. Finally in the late stage, i.e. in the 10th, 11th, and 12th layers, attention is more on prominent patch tokens.

From such attention style, we can conclude the feature extraction and representation mechanism as the Early stage focuses on local and neighboring features, extracting low-level fine-grained features. Then feature representations interact and fuse to generate a strong enough representation in the middle stage. The representation in each token is enhanced by such interaction. And in the late stage, the model focus on the most prominent patch tokens to extract information for final classification.

In future research, it may be possible to only apply head token design in the middle stage of vision transformers to save computation costs. We hope this visualization of the mechanism will inspire more wonderful architectures in the future.

Table 10: Results on 224×224 resolution. All the models are trained from scratch for 100 epochs under the same training schedule.

| Method | #Params | GFLOPs | CIFAR-100 | Clipart | Painting | Sketch |
|---|---|---|---|---|---|---|
| ResNet-50+$\mathcal{L}_{drloc}$ [3] | 21.2M | 3.8 | 72.94 | 63.93 | 53.52 | 59.62 |
| SwinT+$\mathcal{L}_{drloc}$ [3] | 24.1M | 4.3 | 66.23 | 47.47 | 41.86 | 38.55 |
| CvT-13+$\mathcal{L}_{drloc}$ [3] | 19.6M | 4.5 | 74.51 | 60.64 | 55.26 | 57.56 |
| T2T-ViT+$\mathcal{L}_{drloc}$ [3] | 21.2M | 4.8 | 68.03 | 52.36 | 42.78 | 51.95 |
| DHVT-T | 6.0M | 1.2 | 74.78 | 58.94 | 52.64 | 56.66 |
| DHVT-S | 23.7M | 4.7 | **78.64** | **64.75** | **56.42** | **61.35** |

## 5  Results on larger resolution

In order to make a comparison with paper [3], we conduct experiments under the same training scheme. The models are trained from scratch on CIFAR-100, Clipart, Painting and Sketch for 100 epochs and the training resolution is 224×224. The patch size for SwinT, CvT, T2T-ViT and our

DHVT is set to 16. And the following Table 10 demonstrates the performance superiority of our method.

## 6 Model Variants

We present the variants and architecture parameters of our proposed model in this section. Note that all the models remove the absolute positional embedding. For the CIFAR-100 dataset, the image size is 32×32, and for DomainNet and ImageNet it is 224×224. In Table 11, "MLP" represents MLP projection ratio and "S&E" is the reduction ratio in the squeeze-excitation operation.

Table 11: Model variants of DHVT

| Method | Dataset | Patch | DAFF | | #heads | depth | Dim | #Params | GFLOPs |
| | | | MLP | S&E | | | | | |
|--------|---------|-------|-----|-----|--------|-------|-----|---------|--------|
| DHVT-T | CIFAR | 4 | 4 | 4 | 4 | 12 | 192 | 6.0M | 0.4 |
| DHVT-T | CIFAR | 2 | 4 | 4 | 4 | 12 | 192 | 5.8M | 1.4 |
| DHVT-S | CIFAR | 4 | 4 | 4 | 8 | 12 | 384 | 23.4M | 1.5 |
| DHVT-S | CIFAR | 2 | 4 | 4 | 8 | 12 | 384 | 22.8M | 5.6 |
| DHVT-T | Domain | 16 | 4 | 4 | 4 | 12 | 192 | 6.1M | 1.2 |
| DHVT-S | Domain | 16 | 4 | 4 | 6 | 12 | 384 | 23.8M | 4.7 |
| DHVT-T | ImageNet | 16 | 4 | 4 | 3 | 12 | 192 | 6.2M | 1.2 |
| DHVT-S | ImageNet | 16 | 4 | 4 | 6 | 12 | 384 | 24.1M | 4.7 |