# OpenReview forum: "Bridging the Gap Between Vision Transformers and Convolutional Neural Networks on Small Datasets"
_NeurIPS.cc/2022/Conference — NeurIPS 2022 Accept_

### Official Review · Reviewer_4fHM · 2022-07-05

**Rating:** 5
**Confidence:** 4
**Soundness:** 3 good
**Presentation:** 3 good
**Contribution:** 3 good

**Summary:**

This paper focuses on generalizing vision transformers to the domain of small datasets through injecting spatial-wise and channel-wise inductive biases into model architecture design. Specifically, the authors point out that it is hard for the vanilla vision transformers to model spatial relevance and diverse channel representation under data-constrained scenarios. The weaknesses for vision transformers compared to their CNN counterparts finally lead to worse performances on small datasets, *e.g.,* cifar. To address the drawbacks, the authors introduce two augmented modules SOPE and DAFF, corresponding to the patch embedding and feedforward operators. By using additional convolution operations to gather the local feature, spatial relevance within patches is emphasized, and is forced to learn better relations even with limited data. On the other hand, *head tokens* are adopted on top of the widely used classification token and serve as the global token for each group of channels (*i.e.,* a head) in the multi-head self-attention module. In this way, the interaction across different heads is facilitated and shows better representation capability under a low data regime. Regarding the experiments, the authors validate the effectiveness of their module on various datasets, including small datasets like cifar100 and DomainNet, and larger datasets like ImageNet-1K. A good trade-off between model parameters and test accuracy is shown compared to several state-of-the-art baselines, and ablation studies are also conducted to show the contribution of each part.

**Questions:**

Most of my concerns are addressed in the **Strengths And Weaknesses** section. The followings are my detailed questions regarding the weaknesses.

+ I understand that the authors have stated several times in the paper like in the abstract '... the lack of data hinders ViTs to attend the spatial relevance ...' or '... the scarce data can not enable ViTs to learn strong enough representation ...', nevertheless, from my perspective, these statements are more like intuitive claims that short of technique supports. For example, it would be better if some qualitative results can show that spatial relevance is more critical for small datasets. The previous work [1] actually shows that lower attention layers do not learn to attend locally with less training data. Nevertheless, the *less training data* in [1] refers to ImageNet-1k without ImageNet-22k pretraining, which still belongs to large-scale data in the scope of this paper. I wonder if this claim is also correct with small datasets like cifar.

+ Do the setups in table 2 correct? I have checked the wide-resnet paper and found that WRN28-10 is trained for 200 epochs instead of 300 epochs. Also, as I have mentioned before, reporting the best results of 5 runs may enjoy the gain from randomness, especially on small datasets.

+ A comparison on flops-acc trade-off for cifar100 would be better. According to supplementary material, the DHVT-S model that shows the highest performance (85.68) has 6.3G flops, which requires high computation for small image size. The trade-off for computation cost would be more valuable than training parameters.

+ A Typo in line 181: require->required.

[1] Raghu, Maithra, et al. "Do vision transformers see like convolutional neural networks?." Advances in Neural Information Processing Systems 34 (2021): 12116-12128.

**Limitations:**

The authors have addressed the limitations. I feel that inference speed is also a question that is worth further investigation. Although the authors provide the model throughput in the supplementary, the detailed comparison with baseline models and efficient cnns is still lacking.

**Strengths And Weaknesses:**

+ Strengths
  + This paper is organized in an easy-to-follow pattern. Related works on data-efficient vision transformers are thoroughly investigated and properly introduced. Each component of the proposed model is described in detail respectively. The experiments are substantial and some results and experiment details in the supplementary material also help to understand the paper.
  + The practical value of this paper is promising. Although vision transformer models show better performance and scalability on large-scale datasets, the potential application on data constraint scenarios is still under-explored. The gap between vision transformer and cnns on the small dataset is a topic that is worth deep investigation. This paper attempts to address this limitation from the spatial and channel perspective, and the experiments show improvements on widely used benchmarks, *e.g.,* cifar100. I think this shows a new direction to practically using vision transformer models on datasets with low-scale / low-resolution data.
  + The experiments are sufficient in this paper. The parameter-accuracy trade-off on three datasets, ablative studies on each proposed component, visualization on attention maps, and the flops-accuracy trade-off in the supplementary material are detailedly described.
  + Finally, I really approve that the authors are willing to share the code of each module. This really helps to understand the implementation detail.

+ Weaknesses
  + The motivation of this paper can be addressed more clearly. The authors point out that spatial relevance and diverse channel representation are two essential aspects that influence the performance of vision transformers under a low data regime. Nevertheless, I am still concerned about the practical connection between **lack of two inductive biases** with **low data regime**. To my knowledge, the lack of inductive biases is also a question for large-scale datasets like ImageNet-1k. Previous research [1] have already explored strengthening the spatial relationship between patches, and the diversity of features among different heads has also been investigated [2]. As a result, the fact that the lack of two inductive biases leads to inferior performances is not so surprising. As the authors are motivated from these two perspectives to address the limitation of the vision transformer, I think it would be better if they can **emphasize the unique influence of the inductive biases for small datasets**, and explain how the situation differs from large-scale datasets.

  + Technical contribution. It is an interesting idea to improve the performance of the vision transformer from spatial and channel perspectives. However, it must be pointed out that the proposed modules share some resemblance with several previous works. The convolutional patch embedding is widely adopted in several works [1][3]. Using depthwise convolution in the feedforward network is also a common trick [4]. The dynamic aggregation feedforward module also seems like a trivial implementation of SENet, and similar insight has been explored in [5]. It would be better if the authors can focus on addressing the differences with previous works.

  + I also have concerns about some experimental settings. For the main results on cifar100 shown in table 2, I wonder if the comparisons are fair. According to the authors, DeiT augmentations are adopted for proposed models. Do previous cnn models use the same augmentation? Also, the timm implementation can practically improve the model performances of cnn models. Have the authors reimplement the baselines on timm and see the differences with the original paper? I understand it is impractical to ask the authors to re-implement all the baseline approaches, I think a simple comparison with a widely used model like ResNeXt would also do the trick and makes the result more convincing. In section 4.2, the authors state that the results are the best out of 5 runs. This is clearly different from the routine in previous works where an average of 10 runs are usually used for cifar datasets. Considering the large variance for cifar100, I think some part of the gain may come from the randomness.



[1] Xiao, Tete, et al. "Early convolutions help transformers see better." Advances in Neural Information Processing Systems 34 (2021): 30392-30400.

[2] Raghu, Maithra, et al. "Do vision transformers see like convolutional neural networks?." Advances in Neural Information Processing Systems 34 (2021): 12116-12128.

[3] Yuan, Li, et al. "Volo: Vision outlooker for visual recognition." arXiv preprint arXiv:2106.13112 (2021).

[4] Guo, Jianyuan, et al. "Cmt: Convolutional neural networks meet vision transformers." Proceedings of the IEEE/CVF Conference on Computer Vision and Pattern Recognition. 2022.

[5] Yuan, Li, et al. "Tokens-to-token vit: Training vision transformers from scratch on imagenet." Proceedings of the IEEE/CVF International Conference on Computer Vision. 2021.

---

> ### Author Response · Authors · 2022-08-02
> **Response 1 to Reviewer 4fHM**
>
> Thank you for your comments! We hope our comprehensive answers can address you concerns.
>
> **Q1. Influence of inductive biases for small datasets.** Thank you for your suggestion. We think we can provide qualitative result for the spatial relevance. The baseline DeiT-T result is 67.59 on CIFAR-100. And after splitting class token in FFN, as we pointed out in our reply to _Reviewer SHR5_ in Q1, the baseline comes to **68.76**. Now under this circumstance, if we simply introduce an Average Pooling Layer of stride=1 and 3x3 window size without any parameters, the performance will rise to **70.48 (+1.72).** You can see that simply aggregating neighboring feature can help, which means paying more attention on modeling better spatial relevance is of great importance. Due to its special structure and flexibility, ViTs themselves are hard to learn a simple averaging from neighbouring. We need to impose some spatial relevance in helping them training from scratch on small datasets.
>
> Note that **the lack of inductive biases is the intrinsic problem of ViTs, which is amplified on the condition with small datasets.**. The amount of training data just determines whether ViTs can derive a good feature extraction and representation ability itself. We start from the problem of ViTs themselves and provide our modification and evaluate its effectiveness on various size datasets. Previous works have investigated deeply on larger datasets like ImageNet-1K, and in our paper we focus on much smaller datasets. The final goal of our paper is to update the ViTs, making it behave competitive or even better than CNNs. That’s why the title is “Bridging the Gap between CNNs and ViTs”.
>
> When training from scratch on large enough dataset ImageNet-22K, the training data is enough for the ViTs to learn spatial relevance and good channel representation. But when it comes to ImageNet-1K, the intrinsic problem of ViTs gets amplified. The spatial relevance can not automatically learned well and the channel representation maybe not good enough. And considering much smaller CIFAR-100, it is quite a challenge to develop an accurate recognition capability on such scarce data. The spatial relevance is missed, even when the input resolution is small, and the channel representation is worse because of insufficient training data. In conclusion, as the training dataset becoming smaller, the intrinsic problem of lacking inductive biases in ViTs becomes more apparent. So methods that impose strict inductive biases into ViTs is quite essential in training from scratch on small datasets.
>
> **Q2. Comparison with previous works.** We proposed 3 modules accounting for introducing inductive biases of spatial relevance and diverse channel representation. As you pointed out, convolutional patch embedding is widely used in current work and we also adopt it as a simple operation to model spatial relevance. More complex operations like the local window attention in VOLO, as you pointed out, are left for future research. We further introduce two affine transformation in it, enabling a more stable training result. You can also refer to our reply to _Reviewer SHR5_ in Q4.
>
> The DAFF is a combination of an FFN integrated with depth-wise convolution and a dynamic aggregation module for class token. Splitting class token away from patch tokens and passing it identically through FFN is useful and meaningful. You can refer to our reply to _Reviewer SHR5_ in Q1 for the explanation and experiment results. The dynamic aggregation module collects features from patch tokens and re-calibrate class token channel-wise with an SE style. Thanks for your comments, we missed the work CMT and we will put it in the reference in the future version. The previous method CMT also uses DWCONV and a shortcut inside. However, they do not use class token. Here, we need to consider re-calibrate class token. And note that, in this rebuttal time, we provide comprehensive ablation study on the choice of BatchNorm and LayerNorm and their connection between batch size. You can refer to our reply to _Reviewer b3Vm_ in Q6.
>
> VOLO uses a fine-grained patch embedding module with attention in local window  but it is sophisticated and brings huge computational burden. We just apply a convolutional patch embedding for simplicity. And compared T2T-ViT, which leverages SE right after multi-head self-attention, it is far more different from our method. It is somewhat similar to the function of our Head-Interacted Multi-head Self-Attention, which enables different head interact with each other for better channel representation. The SE in T2T-ViT is a channel-wise post-processing, re-calibrating each channel. However, **our Head Token is group wise, dividing channels into several groups for simultaneous processing, which is more compatible with multi-head mechanism**.

---

> > ### Author Response · Authors · 2022-08-02
> > **Response 2 to Reviewer 4fHM**
> >
> > Owing to the limited characters of only one response and we want to anwer your questions in details, we have to add another reply!
> >
> > **Q3. Experiment Settings and More Results.** Thank you for your great suggestion! We implement **ResNeXt50-32x4d**, which has **23.7M** parameters on DomainNets and is comparable with our proposed DHVT-S. And also suggest by Reviewer u2nu, we implement **CvT-13** also. However, **because of the limited computation resource and tight rebuttal time, we can only train them on some of the datasets**. The results are as follows. The results show that even compared with ResNeXt, our method is still superior, demonstrating the effectiveness of our method. The comparision of ResNet50, ResNeXt50-32x4d, CvT-13 and our DHVT are summaried in the following table.
> >
> > |Method| #params |ClipArt|Painting|Sketch|Infograph|Real|Quickdraw
> > |:-------:|:--:|:--:|:--:|:--:|:--:|:--:|:--:|
> > |ResNet50| 24.2M |71.90|64.36|67.45|32.40|81.51|74.19|
> > |ResNeXt50-32x4d| 23.7M |72.93|64.37|68.52|-|-|-|
> > |CvT-13|19.7M|69.77|61.57|66.16|-|-|-|
> > |DHVT-T|6.1M|71.73|63.34|66.60|-|-|-|
> > |DHVT-S|23.8M|**73.89**|**66.08**|**68.72**|**35.11**|**83.64**|**74.38**|
> >
> >
> > On comparison with CNNs on CIFAR-100, we are not under the same data-augmentation. Some of the augmentations are suitable for ViTs but not for CNNs, which may hinder the performance of CNNs. So we just cite the reported highest results from the corresponding paper. And we also try to re-implement ResNeXt50-32x4d on the CIFAR-100 under the same augmentation and training method. However, it only achieves 60.20 accuracy. We hypothesize that maybe some data augmentation is not suitable for CNNs under such low-resolution 32x32 input.
> >
> > Here we train 5 more results with different random seed. The follows are total 10 runs results for our DHVT-S with patch size of 4 on CIFAR-100: {82.91, 82.77, 82.82, 82.86, 82.88, 82.94, 82.75, 83.02, 82.79, 83.00}, and the average result is 82.87±0.08.
> >
> > We are trying our best to filling up the comparison and promise to report it on the final version.
> >
> > **Q4. Incorrect writings.** We are sorry that we made a mistake on the training epochs of WideResNet! We will fix it in the future version. And we will also correct the wrong writings as you pointed out.
> >
> > **Q5. Flop-Acc Trade-off.** We admit that this is the limitation of our proposed method. We conduct comprehensive computation under small amount of parameters. The dynamic operations in our method is responsible for the high computation burden. However, it can be simplified. Note that Head Tokens receives more attention in the input layer, middle and final layers, rather than shallow layers, which can be seen from Fig 7.  in the Supplementary Material. We hypothesis that Head Token is more useful when processing high-level semantic features. Thus maybe only applying Head Tokens in the deeper layers of the backbone will not hinder the performance too much. Under this concern, the computational cost can be reduced.

---

> > > ### Comment · Reviewer_4fHM · 2022-08-08
> > > **Response to the authors**
> > >
> > > I would like to thank the authors for their detailed responses. Most of my concerns are addressed and I believe this is an interesting touch towards promoting vit performances on small-scale datasets. In general, I believe the pros outweight cons and I would keep my score unchanged.

---

> > > > ### Author Response · Authors · 2022-08-08
> > > > **Post-rebuttal to Reviewer 4fHM**
> > > >
> > > > We would thank you for your detailed comments and the suggestions your raised help us a lot to improve this work.
> > > >
> > > > Yet, it would be a little bit pity that we could not further improve our status in the response. Sincerely, could you please let us know whether there is any question we have not addressed properly. We humbly seek your further advice to improve this work and cherish the possible opportunity during this discussion period.
> > > >
> > > > Thanks again for your time and help.

---

### Official Review · Reviewer_SHR5 · 2022-07-09

**Rating:** 4
**Confidence:** 4
**Soundness:** 3 good
**Presentation:** 2 fair
**Contribution:** 2 fair

**Summary:**

This paper proposes a new Vision Transformer structure named DHVT targeting to improve the performance of ViTs on the image classification task with small datasets. Specifically, DHVT is designed to improve the spatial relevance among the patches with convolutional inductive bias and enhance the channel representation by re-calibrating the channel representations. To validate the effectiveness of the design choices, the authors conducted experiments on small datasets represented by CIFAR-100 and ImageNet-1k. The CIFAR-100 results look good, and the proposed method outperforms the other baseline methods by large margins.

**Questions:**


Please address the issues pointed out in the weakness.

**Limitations:**

In general, yes.

**Strengths And Weaknesses:**

Strengths:
- It’s meaningful to study why and how ViTs are inferior to CNNs when training on small datasets from scratch.
- The quantitative results in Table 2 look good and surpass the other methods.

Weakness:
- It’s hard to understand the motivation for the proposed detailed design choices and why they are better than other simple designs. For instance, why re-calibrating only the class token with SE operation [25], considering localViT [18] proposed re-scaling channels for all tokens in FFN blocks? A proper discussion with the SE module in localViT is needed. Also, the head tokens are proposed to enhance the channel groups for each head by modeling head correlations, which can easily be implemented by an MLP layer. Can you explain the benefit of modeling the head correlations with MSA instead of a simple MLP layer?

- The experiments are insufficient to support the authors’ claim. Tables 3 and 4 have very few baseline methods and it’s hard to see the gain of DHVT over the SOTA methods.

- The novelty for modeling spatial relevance is limited, since introducing early convolutions has been shown effective in previous works, e.g. [a].

[a] Xiao, Tete, et al. "Early convolutions help transformers see better." Advances in Neural Information Processing Systems 34 (2021): 30392-30400.

---

> ### Author Response · Authors · 2022-08-02
> **Response to Reviewer SHR5**
>
> Thank you for your comments! We hope our comprehensive answers can address you concerns.
>
> **Q1. Re-calibrating only class token.** This is a good point. The feed forward network (FFN) is responsible for non-linearly projection and re-calibrating the tokens themselves. Here we introduce Depth-wise Convolution (DWCONV) into FFN, gathering neighboring feature inside FFN. Note that DWCONV can only be operated on patch tokens, and so class token is passed identically through. Now we also want to re-calibrate the class token and we want to use the information from patch tokens. Thus, we gather the feature of all the patch tokens after their projection and re-calibrate class token in an SE style. And we also find that, **splitting class token away from patch tokens  is helpful** and we list the ablation study as follows.
>
> (1) **Baseline results** of ViT training from scratch on CIFAR-100 is **67.59**.
>
> (2) On original ViT, if we split class token away from patch tokens, passing identically through FFN, the result will be **68.76 (+1.17)**.
>
> (3) Under (2), if we re-calibrate class token using the method in our paper, the result will be **69.34 (+1.75).**
>
> From (1)-(3), we conclude that **class token should NOT be projected and re-calibrated in the same way as patch tokens**. We argue that **class token is responsible for collecting information, and its representation is different from patch tokens**. If both class token and patch tokens are processed together by FFN, the model will be confused because the two kinds of tokens are quite different intrinsically.
>
> As you mentioned, SE is also applied in LocalViT. So we provide our discussion here. In LocalViT, SE operation is used right after h-swish. They are adopted after in between two linear layers, right after DWCONV. So we point out that it is served as an activation function to some extent. However, in our paper, we use SE operation to gather information from patch tokens and re-calibrate class token. The intuition is quite different. And we conduct experiment to show that **if we apply SE to re-calibrate on all the tokens, instead of only class token, the performance will drop from 80.98 to 78.34**, which greatly decrease the performance.
>
> **Q2. Head Token v.s. Simple MLP.** There are two benefits. **First,** compared with a simple MLP which stays fixed after training, using Head Token to enhance interaction among channels is more dynamic and data-specific. Owing to the scarce training data of small dataset, more comprehensively and data-specifically method can make full use of the data. **Second,** a simple MLP is point-wise on each channel. It only projects each channel respectively, rather than considering a group of channels as a whole. In our design, the Head Token is compatible with the Multi-head mechanism, which calculates relationship using a group of channels from input data. Previous works did not consider channels from the view of group and we are the first to provide this kind of idea.
>
> **Q3. Insufficient Baselines.** Previous CNN works did not train their models on DomainNet. So in our main paper we implement ResNet50 as the baseline. And for the ImageNet-1K, the number of pages is limited so we have to put the detailed comparison in supplementary materials. In the rebuttal period, we implement **ResNeXt50-32x4d** and **CvT-13** as baselines on DomainNets. Please refer to *Reviewer 4fHM* in Q3 for comprehensive results. For the comparisons on ImageNet-1K datasets, please refer to Table 4 in Supplementary Materials! We provide enough comparison there.
>
> **Q4. Spatial Relevance.** Our SOPE and DAFF provides inductive biases on spatial aspect. They are modified from previous works. As you point out, we follow [a] to conduct Patch Embedding in a sequential style, and it is indeed effective. We also provide our modification that we introduce two affine transformation before and after the sequential convolution layers. The pre-affine is done on transforming the original input and the post-affine is to adjust the feature after convolution sequence. This method renders a stable training results on CIFAR-100 dataset. If we remove such operation, the **average performance** **will drop from 80.90 to 80.72**. On the Vanilla ViT with SOPE, the performance is **73.68**, and if removing the two affine transformation, the accuracy will drop to **73.42**.
>
> And for DAFF, we introduce a shortcut alongside the DWCONV. **If we remove the shortcut, the result will drop from 80.98 to 80.14**, demonstrating the effectiveness of this dynamic operation. The BatchNorm is also adopted after the convolutions. Thank to the suggestion by _Reviewer b3Vm_ and _Reviewer u2nu,_ we conduct comprehensive experiments to show that the choice of BatchNorm instead of LayerNorm can bring higher performance while somewhat sensitive to the batch size. You can refer to _Reviewer b3Vm_ in Q6 for the results.

---

> > ### Author Response · Authors · 2022-08-09
> > **Post-rebuttal Response to Reviewer SHR5**
> >
> > Thank you for your detailed comments again and the questions you raised are full of insight which helps us rethink and improve our work a lot.
> >
> > Nevertheless, we are not sure that if our response is able to address your concerns properly. Sincerely, if you have further questions or if our response on some points is not precise, please let us know.
> >
> > We are looking forward to your further reply and thanks for your time and help.

---

> > ### Comment · Reviewer_SHR5 · 2022-08-09
> > **Responses to Authors' responses**
> >
> > I have read through the authors' responses and read other reviewers' comments as well.
> >
> > I appreciate the detailed responses from the authors. It does explain some details while some of my major concerns still remain:
> >
> > - As shared with other reviewers, the novelty is limited. Although the authors point out the detailed differences, the general ideas of spatial relevance and diverse channel representation are quite common. For the claimed three contributions, the first one is mentioning the overall solution containing the two components, and the 2nd and 3rd ones basically detail the two-component contributions. Although I agree there are some deltas or new tricks, I didn't see a significant one.
> >
> > - Although the authors provide more comparisons for Tables 3 and 4, the results are not presented in a consistent way. Table 2 includes Patch Size and Epochs columns with a lot of comparing methods, while Tables 3 and 4 only contain Params and Accuracy, and Table 4 also has GFLOPs information. It is hard to clearly see the accuracy-parameter-complexity tradeoff and the comparisons with SOTA across multiple datasets. Moreover, we can see that the Res2NeXt-29, 6c×24w×6s-SE is the most competitive one for CIFAR-100, while its results are not reported in Table 3&4.
> >
> > A minor issue: Although it is ok to use multiple responses at one time to address one reviewer' comments in detail, it might not be fair to others.

---

> > > ### Author Response · Authors · 2022-08-09
> > > **Post-response 1 to Reviewer SHR5**
> > >
> > > Thank you for your responses and your time again. We aim at replying to you more detailed and sincerely and so we have to use multiple responses. We hope this two detailed post-reponses for you could address your concerns.
> > >
> > > **Q1: Novelties.** As you pointed out, spatial relevance and diverse channel representation are two common points on promoting Vision Transformers. However, previous works only achieve data-efficiency on medium dataset ImageNet-1K. **And not all the modification made by previous works can truly bridge the performance gap between CNNs and ViTs on such small datasets like CIFAR-100.** As you can see in paper [1] Table 4, one of the paper we referred to, Swin Transformer, T2T-ViT and CvT still remain performance gap or just be comparable to the  ResNet-50. Though they successfully reach promising results on ImageNet-1K, they fail to compete with or just behaves equally to the most common CNN ResNet-50. While in our work, the DHVT we proposed is able to close the gap between common CNNs like ResNet and ResNeXt, and we can even defeat much stronger CNNs like SKNet, DenseNet and Res2Net. **Our method exhibits superior performance on a wide range of datasets**, CIFAR-100, DomainNet and ImageNet-1K, demonstrating the generalization of our method. **Our modification is more effective on solving the intrinsic problem of ViTs, so the good generalization capacity is not surprising.**
> > >
> > > And as we pointed out in the first-round response to you in Q2 and the first-round  response to *Reviewer b3Vm* in Q5, the highlighted Head Token is a brand new design. It brings a new insight that **enhancing channnel representation from the view of groups rather than independent ones.** The Figure 7 in our Supplementary Materials reveals the feature interaction pattern in our work. We further raised a question that if such characteristic is general in other Vision Transformers. We hope this will inspire more future research.

---

> > > > ### Author Response · Authors · 2022-08-09
> > > > **Post-response 2 to Reviewer SHR5**
> > > >
> > > > **Q2. Experiment Comparisons.** We are sorry that it is our fault that the results are not presented in a consistent way. We will make it more clear in the future version. The reasons are as follows.
> > > >
> > > > **First,** the page limitation does not enable us to present all the results in the same way, and have to show the detailed comparison in the Supplementary Materials. Because CIFAR-100 is the main target to evaluate our method, so we present the whole comparison in the main paper.
> > > >
> > > > **Second, when compared with baselines on CIFAR-100**, we do not show the computational complexity of baselines and we only put the measurement of our method in the Supplementary Materials. **The results from the CNN baselines also do not show their computational complexity**, like in Res2Net [2] Table 4 and DenseNet [3] Table 2, SKNet [4] in Table 5, and their model variants are also not provided in the code. Res2NeXt-29, 6cx24wx6s-SE is more like a model variant specially for CIFAR-100. **So for the consistent comparison format in the baselines works, we just show the number of parameters to measure the model complexity.** On compared with ViT baselines on CIFAR-100, we report the patch size which is an important parameter in ViTs. The patch size choice also influences the performance of our method. When it comes to Epochs, the CCT [5] trains on different epochs. It reports results with training epochs of 300, 500 and 1500 and we have to show that which one is choosen to be our baseline. **And for fair comparison, the epoch in our work on CIFAR-100 is set to 300, which is the same as other ViTs and CNNs baseline except for wide-resnet of 200 epochs.** So considering the factors above, in the table of CIFAR-100 results, we show the number of parameters, epochs and patch size.
> > > >
> > > > **Third, in the comparisons on DomainNet,** previous work except paper [1] do not implement their model on DomainNet datasets. So we implement ResNet50 in the main paper and ResNeXt50-32x4d in the rebuttal period as the baselines on DomainNet. These two network are the common model variant choice. While Res2NeXt-29, 6c×24w×6s-SE is not a common choice even in their main paper, so we do not implement it on DomainNet datasets. And we report the number of parameters as the model complexity. Note that the patch size in our work and CvT, which is reported in our first-round response to *Reviewer 4fHM*, is set to 16 and the epochs of all models are 300. So we did not show the patch size and epochs in the Table 3, and we show the experimental setup in details in the text part.
> > > >
> > > > **Fourth, the format of comparision on ImageNet-1K is consistent to the previous ViT works** and most of them report the number of parameters, FLOPs and training resolution. In this  part we mainly compare with ViTs, so we just cite the results of two latest CNN RegNet and ConvNeXt.
> > > >
> > > > Thanks again for pointing out our shortages and we will follow your suggestion to update the comparison tables in the future version.
> > > >
> > > >
> > > > [1] Liu Y, Sangineto E, Bi W, et al. Efficient training of visual transformers with small datasets[J]. Advances in Neural Information Processing Systems, 2021, 34: 23818-23830.
> > > >
> > > > [2] Gao S H, Cheng M M, Zhao K, et al. Res2net: A new multi-scale backbone architecture[J]. IEEE transactions on pattern analysis and machine intelligence, 2019, 43(2): 652-662.
> > > >
> > > > [3] Huang G, Liu Z, Van Der Maaten L, et al. Densely connected convolutional networks[C]//Proceedings of the IEEE conference on computer vision and pattern recognition. 2017: 4700-4708.
> > > >
> > > > [4] Li X, Wang W, Hu X, et al. Selective kernel networks[C]//Proceedings of the IEEE/CVF conference on computer vision and pattern recognition. 2019: 510-519.
> > > >
> > > > [5] Hassani A, Walton S, Shah N, et al. Escaping the big data paradigm with compact transformers[J]. arXiv preprint arXiv:2104.05704, 2021.

---

### Official Review · Reviewer_u2nu · 2022-07-11

**Rating:** 6
**Confidence:** 5
**Soundness:** 2 fair
**Presentation:** 2 fair
**Contribution:** 2 fair

**Summary:**

The paper proposes a new Vision Transformer architecture that merges CNNs and ViT to have the best of the two worlds. The paper focuses on small datasets and Vision Transformers, which have to be somehow regularised or helped to learn the inductive biases that are naturally enforced with CNNs.

The paper proposes a new architecture that modifies the FFN with a Dynamic Aggregation Feed Forward, and proposes a "Head Token" so that each head of the MHA can attend to all others somehow. This supposedly helps learning better features.

The authors experimented the proposals with various small datasets and observed improvements in all tests.

**Questions:**

- The Transformer is a good architecture also because it does not depend too much on the batch size. However, your architecture now has Batch Normalization. Why? Couldn't you use Layer Norm? If not, why?

- Why didn't you compare with [54]?

- What is the impact of the batch size on your results?

- what if I use this architecture with a big dataset? Is this an architecture that is general or only for small datasets? what is your opinion here

- What about downstream tasks? Does your architecture help there?

- answer the other questions raised in the previous section plz!

**Limitations:**

No limitations

**Strengths And Weaknesses:**

PROS:
- the problem is very important: Vision Transformers focus a lot on big datasets but small datasets are nowadays almost forgotten
- the paper is clear

CONS:
- lack of comparison with some state of the art
- lack of clarity on the motivations for some choices

DETAILS:
To me some parts of the paper are unclear. E.g.:
- the choice of batch norm
- why you did not compare with CvT, very famous architecture joining CNNs with Transformers
- why you did not compare with [54], published one year ago in the same conf
- why is the Head Token beneficial? I mean, the multi-head attention was created to have somehow the same approach CNNs have with multiple filters: learning different aspects of the same input. I am surprised your modification helps

I did not also like the lack of clarity of the experiments. Why is the resolution of CIFAR-10 changed? What happens if you do not do the changes explained in L244? Are all baselines trained with the original augmentations, params, resolutions?

Then, [54] reports 75.22 on ResNet-50 on ClipAart, 66.58 on Paintings etc. Your results are lower both for the baseline and your method. Why?

L288 why 4 heads? it is not standard at all.

L265 DomainNet is much bigger. Why did you select only a few datasets from DomainNet? Unfortunatedly, this raises the question on me whether you cherrypicked the results. Moreover, depending on the Table (2, 3, 4) the baselines change. The reason is a bit unclear.


OTHER:
- L144: needs a citation, e.g. 54. However, 54 shows that the networks are competitive with ResNet-50
- L55-79: it is not clear why it is important to solve everything in the spatial dimension.
- L79 SE is not defined
- L150: non-hierarchical here is not clear. Try to explain it better.
- L180: this is a result I suppose, on the methods. If you want to state this, show the results.
- L181: more solution is not english
-

---

> ### Author Response · Authors · 2022-08-02
> **Response 1 to Reviewer u2nu**
>
> Thank you for your comments! We hope our comprehensive answers can address you concerns.
>
> **Q1. BatchNorm/LayerNorm/The Influence of batchsize?** Please refer to _Reviewer b3Vm in Q6._We provide very detailed experiment results there.
>
> **Q2.Comparison with CvT.** Following your suggestion, we re-implement CvT-13, which has 19.9M parameters. Owing to the limit computation resource and tight rebuttal time, we can just train CvT-13 on CIFAR-100, ClipArt, Painting and Sketch  under the same experiment setup as ours. The result is as follows. Note that CvT-13 a hierarchical structure which has 3 stages. We keep the patch embedding stride=1 in the 2nd and 3rd stages, which means the 2nd and 3rd stages do not conduct downsampling. So here the patch size for CvT is compatible with the patch embedding stride of the 1st stage. If we keep the original patch size of 16 on CvT-13, where patch embedding strides for the 3 stages are 4, 2, 2, training from scratch on CIFAR-100, the results will only be 67.56, showing that large patch size is not suitable for small resolution input data.
> | Method | Resolution |Patch size |CIFAR-100 |
> |:--:|:--:|:--:|:--:|
> | CvT-13 | 32 |4 | 79.24 |
> | DHVT-T (Ours) |32 | 4 | 80.93 |
> | DHVT-S (Ours) |32| 4 | 82.91 |
> | CvT-13 |32| 2 | 81.81 |
> | DHVT-T (Ours) |32| 2 | 83.54 |
> | DHVT-S (Ours) |32| 2 | 85.68 |
>
> And for the **DomainNet datasets**, the results are:
>
> | Method | Resolution |Patch size | ClipArt | Painting | Sketch |
> |:--:|:--:|:--:|:--:|:--:|:--:|
> | CvT-13 | 224 |16 | 69.77 | 61.57 | 66.16 |
> | DHVT-T (Ours) | 224 |16 | 71.73 | 63.34 | 66.60
> | DHVT-S (Ours) | 224 |16 | 73.89 | 66.08 | 68.72
>
>
> **Q3. Compared with [54]/Change of resolution.** Though both of our goals are training from scratch with ViTs, **the experimental setting is quite different**. Note that the original resolution of CIFAR-100 dataset is 32x32. Paper [54] trains for 100 epochs and **they resize the CIFAR-100 from original 32x32 resolution to 224x224**, while our setting is training for 300 epochs and **we keep the original resolution 32x32**. There are two reasons. **First,** training for longer epochs is more close to performance convergence, which is also presented in [54] with 300 epochs results rather than just 100. **Second,** keeping original resolution is consistent with the experiment setups of previous CNNs and ViTs work. Our experimental setup is quite the same as NesT [23].
>
> When compared with CNNs on CIFAR-100, we just cite the reported result in their respective papers, because CNNs may not works well on such strong data-augmentation in ViTs (previous work have pointed out also). We re-implement ResNeXt50-32x4d, training with the same setting as ours on CIFAR-100, the performance is only 60.20%! It also shows that CNNs methods have their own suitable data-augmentation. We can not just put them under the same training setting directly. But when training on DomainNet dataset, where the input resolution is 224x224, both of our method and ResNet-50 are under the same training setting, and we also re-implement ResNeXt50-32x4d on DomainNet. You can refer to *Reviewer 4fHM* in Q3.
>
> **Q4. Why [54] reports higher results?** Note that **75.22 on ClipArt and 66.58 on Painting you pointed out is the fine-tuning results, not the train-from-scratch result!** We are talking about train-from-scratch result in the main paper in Table 3. Since paper [54] only train-from-scratch ResNet-50 for 100 epochs on DomainNet, we can not just cite their results. Under the same setting, we re-implement ResNet50 and train it from scratch for 300 epochs.  **We also re-implement ResNeXt50-32x4d on DomainNet in rebuttal period**. You can refer to *Reviewer 4fHM* Q3 for detailed results. The results also show the superiority of our method when training from scratch. **If you are interested in fine-tuning results, you can refer to Table 3. in the Appendix.** The experiments are conducted under the same fine-tuning setting as paper [54]. You can see that our performance is much better than the result reported in [54].
>
> **Q5. The number of heads.** It is well-known that DeiT-Tiny and DeiT-Small have 3 and 6 head respectively. However, this parameter is chosen by the experiment on ImageNet. Similarly, we set the number of heads as 4 and 8 for our DHVT-T and DHVT-S based on the experiment on CIFAR-100. For **vanilla ViT** on CIFAR-100, using **3 and 4 heads will get 66.50 and 67.59** repectively. And the **DHVT-T with 3 and 4 head** will achieve **80.92 and 80.98** respectively. If we choose **4, 6, and 8 heads in DHVT-S, the performance will be 82.27, 82.59, 82.82. So 8 heads is best in DHVT-S and 4 for DHVT-T**.  In compatible with scalibility, we adopt 4 heads in DHVT-T and 8 heads in DHVT-S. We hypothesize this is because each attribute of the object in CIFAR does not need too many channels for representation. So we keep the number of channels in each head less than usual.

---

> > ### Author Response · Authors · 2022-08-02
> > **Response 2 to Reviewer u2nu**
> >
> > Owing to the limited characters of only one response and we want to anwer your questions in details, we have to add another reply!
> >
> > **Q6. DomainNet and baslines.** DomainNet has 6 different dataset as we know, and we show the results on 3 of them. The Quickdraw and Real datasets has training size of 120750 and 120906, and the classes is still 345. The average number of images per class comes to 350, which is quite larger than ClipArt, Painting and Sketch. It does not mean we did not evaluate our methods on these datasets. We have a comparable targer dataset CIFAR-100 now, in which the low-resolution makes training even harder. So results on Quickdraw and Real are not shown in main paper.
> >
> > And for Infograph dataset, the images and the corresponding labels are quite weird. You can investigate the dataset in details and you will find that: For example, many very long images that with a very small peanut is labeled as Peanut! However, there are many other fruits and vegetables in that images, which greatly hinders model training. **So, we think this dataset has its intrinsic problem between images and labels, resulting a very bad dataset**. You can see from paper [54], they also reported very bad results on Infograph dataset. So we give up this dataset.
> >
> > **To show that we do NOT pick the good results purposely**, in the rebuttal period, we evaluate our methods and ResNet50 on the remaining datasets Real, Infograph and Quickdraw. Our **DHVT-S** has **83.64, 35.11 and 74.38** on Real and Infograph respectively, while **ResNet50** only achieves **81.51, 32.40 and 74.19** accuracy. This greatly supports our good method. The whole comparison are summarized in our reply to *Reviewer 4fHM* in Q3.
> >
> > |Method|Infograph|Real|Quickdraw|
> > |:-------:|:--:|:--:|:--:|
> > |ResNet50| 32.40|81.51|74.19|
> > |DHVT-S|**35.11**|**83.64**|**74.38**|
> >
> > On considering baseline changes, the reason is that: We need enough previous work as reference, and the **main dataset** to evaluate our method is CIFAR-100. Fortunately, many previous CNNs and ViTs conduct experiments on CIFAR-100, so we have enough baselines. However, except paper [54], we do not find any other works that train-from-scratch on the DomainNet datasets, and as we said above, the different experiment settings do not enable us to directly cite the results from [54]. So in our paper, we re-implement ResNet50 as baseline and we provide results of ResNeXt50-32x4d in rebuttal period. You can refer to _Reviewer 4fHM_ in Q3. And the ImageNet-1K is widely investigated so we have many baseline methods, and we just cite their results.
> >
> > **Q7. Solving things on spatial dimension.** This is a good point. We think that spatial dimension is the basis of good performance. A good method on spatial dimension means the model can correctly choose which position to focus. Under the correct spatial focus, the re-calibration on channel representation is meaningful and more helpful. So, the problem on spatial dimension must be solved.
> >
> > **Q8. Non-hierarchical.** The non-hierarchical structure means that every encoder block shares the same parameter setting, processing the same shape of features, such as vanilla ViT, CeiT, LocalViT. They do not down-sample and increase channel dimension as the layer goes deeper. While hierarchical structure, such as PVT, SwinT and PiT,  is similar to CNNs style, with smaller resolution and more dimension as going deeper.
> >
> > **Q9. The convergence between CNNs and Ours.** Here we show the convergence curve of max accuracy **every 20 epochs of the total 300 epochs**. The models are DHVT-S and ResNet50, training from scratch on ClipArt datasets. This trend that our method converges faster than ResNet is general among various datasets.
> >
> > DHVT-S: {19.06, 44.27, 56.65, 62.74, 66.48, 68.10, 69.94, 71.17, 71.95, 72.58, 73.23, 73.56, 73.73, 73.85, 73.89}
> >
> > ResNet-50: {8.46, 20.58, 35.89, 48.49, 56.80, 60.92, 63.99, 66.37, 68.36, 69.37, 70.53, 71.37, 71.62, 71.75, 71.90}
> >
> > **Q10. Incorrect writings.** Thanks for your detailed comments! We will fix them in the future version.
> >
> > **Q11. Why is Head Tokens beneficial.** This is a good question! Please refer to our reply to _Reviewer b3Vm_ in Q3. We provide comprehensive explanation there.
> >
> > **Q12. On big dataset.** Please refer to our reply to _Reviewer b3Vm_ in Q4 and *Reviewer 4fHM* in Q1.
> >
> > **Q13. Downstream tasks**. Thanks for suggestion. However, due to the limited computation resources and tight rebuttal time, we could not provide results on downstream tasks and it is left for the future. We have provided fine-tuning results in the Appendix and you can refer to Table 1 and 3 in the Appendix.

---

> > > ### Comment · Reviewer_u2nu · 2022-08-08
> > > **thank you**
> > >
> > > I thank the authors for the very detailed rebuttal. I appreciated it. However, I am still not convinced by BN, which depends on the batch size. Moreover, even if 54 uses another resolution, I think it is crucial to compare the two approaches at both resolution (original vs 224)

---

> > > > ### Author Response · Authors · 2022-08-09
> > > > **Post-response to Reviewer u2nu**
> > > >
> > > > Thanks for your time and comments again. Your suggestions and insights help us rethink our work and make it more solid. Here in this post-rebuttal response, we provide further explanations and we hope this time our responese is convincing enough.
> > > >
> > > > **Q1: Further on LN and BN.** Thanks for your further concern on BN and LN. As we have shown in the first-round response, using BN can reach higher accuracy while it is more sensitive to batch size. From our point of view, the vanilla vision transformer adopts LN before MHSA and MLP, aiming at regularizing each token on channel dimension. This is important because MHSA use dot-product operation, and LN helps control the value of query and key, avoiding extreme values. While in our work, LN is also adopted at the same place, before MHSA and MLP. **Further, we use depth-wise convolution operation, and its output should be regularized in terms of spatial dimension.** When we replace BN with LN, the feature will be reshape into sequence style, which may ignore the spatial relations. **So it is more suitable to use BN for depth-wise convolution for higher recognition accuracy**. If the computational resource is limited and researchers could not search for the best choice of batch size, we think some special BN variants such as Cross-Iteration Batch Normalization [1] can be adopted to reduce the influence brought by small batch size. And there could still hides room for improvement from BN with further computational resource.
> > > >
> > > > We will release all the code, including training and the models, as soon as our work is accepted. So the fairness of all our experiments and the effectiveness of our method are ensured.
> > > >
> > > > [1] Yao Z, Cao Y, Zheng S, et al. Cross-iteration batch normalization[C]//Proceedings of the IEEE/CVF Conference on Computer Vision and Pattern Recognition. 2021: 12331-12340.
> > > >
> > > >
> > > > **Q2:  Training on 224 resolution.** Following your post-response in this discussion period, we implement our method on 224 resolution on CIFAR-100, ClipArt, Painting and Sketch. Note that we keep the same training scheme of paper [54], where the training epoch is set to 100 and the patch size is 16. The baseline models are cited from [54], training with its proposed method. The results are as follows.
> > > >
> > > > |Method|Epoch|Resolution|Patch Size|CIFAR-100|ClipArt|Painting|Sketch|
> > > > |:-------:|:--:|:--:|:--:|:--:|:--:|:--:|:--:|
> > > > |ResNet50| 100 |224|1|72.94|63.93|53.52|59.62|
> > > > |SwinT| 100 |224|16|66.23|47.47|41.86|38.55|
> > > > |CvT-13|100|224|16|74.51|60.64|55.26|57.56|
> > > > |T2T-ViT|100|224|16|68.03|52.36|42.78|51.95|
> > > > |DHVT-T(Ours)|100|224|16|74.78|58.94|52.64|56.66|
> > > > |DHVT-S(Ours)|100|224|16|**78.64**|**64.75**|**56.42**|**61.35**|

---

### Official Review · Reviewer_b3Vm · 2022-07-11

**Rating:** 5
**Confidence:** 4
**Soundness:** 3 good
**Presentation:** 3 good
**Contribution:** 2 fair

**Summary:**

The paper proposes Dynamic Hybrid Vision Transformer (DHVT) for visual recognition. They argue that spatial relevance and diverse channel representation are two important inductive biases for visual recognition and they address the issue from two perspectives.
1. On the spatial aspect, they adopt a hybrid structure, they integrate convolution into patch embedding and multi-layer perceptron (MLP) module, forcing the model to capture the token features and their neighbouring features.
2. On the channel aspect, they introduce a dynamic feature aggregation module in MLP and head tokens in multi-head self-attention module to help re-calibrate channel representation and make different channel group representation interacts with each other.
The experiments are conducted on CIFAR-100, ClipArt, Sketch, Painting, and ImageNet. All of these datasets are for classification.

**Questions:**

See the Strengths And Weaknesses part.

**Limitations:**

See the Strengths And Weaknesses part.

**Strengths And Weaknesses:**

Pros:
- Well-organized and easy to read.
- Motivation is clear from both the spatial and channel sides.
- Good results on many small- and medium-size datasets.
- Nice ablation study is shown in Tab. 5.
- LImitation is stated.

Cons:
- Some unclear captions should be more concrete. E.g., DAFF in Figure. 1.
- Limited novelties. The depth-wise convolution is well investigated in previous works, like CVT, CPVT, PVT v2, etc. And the use of both channel-wise and spatial-wise representation is explored in DaVIT, CoaT, etc.
- What is the intuitive explanation of the HI-MHSA? The component is a bit complicated.
- Why is the work suitable for small datasets? Why does the author put the experiment on ImageNet in Appendix? Which part is specifically designed for small datasets?
- No downstream tasks were evaluated. It would be better if the authors could provide some results on ADE20K or COCO.
- The use of batchnorm, how does it influence the performance? VIT-based architectures typically use layernorm so that the batch size will not hinder the performance.

---

> ### Author Response · Authors · 2022-08-02
> **Response 1 to Reviewer b3Vm**
>
> Thank you for your comments! We hope our comprehensive answers can address you concerns.
>
> **Q1.The limited novelties.** In this paper, we provide 3 components, SOPE, DAFF and HI-MHSA, which aim to overcome the inductive bias problmes of ViT with spatial relevance and diverse channel representation. Some minor operation in our proposed method are also evaluated by other works such as depth-wise convolution (DWCONV). As is also pointed out by _Reviewer 4fHM_, using DWCONV is a common trick in current feed forward network (FFN) design.  It is now becoming a kind of designing paradigm. Here, standing on the shoulder of giants, we also adopt such paradigm hybrid architecture and we evaluate its effectiveness on small datasets. Nevertheless, we provide our modification and it is able to comprehensively leverage the scarce training data with our proposed dynamic modules.
>
> Our DAFF is dynamic on two aspects. **First**, it is dynamic spatial-wise. After integrated with DWCONV, the FFN can be seen as a series of convolution layers. We add a shortcut alongside the DWCONV, resulting a data-specific representation of the input data. Without this shortcut, the performance of DHVT-T will drop from 80.98 to 78.34. **Second**, it is dynamic channel-wise. After feature representation process by convolution-based FFN, we add a dynamic aggregation module special for class token. DWCONV is only adopted for patch tokens, and so class token is passed identically through the FFN. **We aim at aggregating patch token features to re-calibrate class token, enabling the class token to adjust itself**. You can refer to *Reviewer SHR5* in Q1 for detailed experiment results.
>
> Though the previous work like CoaT and DaViT try to enhance spatial-wise and channel-wise representation simultaneously, we enhance channel-wise representation in a different way. They achieve channel-wise representation through conducting self-attention on transposed tokens, while we introduce Head Token together with normal patch tokens and CLS token to efficiently modeling relationship spatial-wise and channel-wise. **The computation of Head Token is light-weight and it does not need to re-formulate the original spatial-wise MHSA**.
>
> **Q2.Downstream Tasks.** Thanks for suggestions. Our main goal is to on the image classification task and we provide fine-tuning results in the Supplementary Materials as Table 1 and Table 3. Due to the limited computation resource and tight rebuttal time, we could not provide results on COCO and ADE20K. We are confident about our methods and the experiments are left for the future version.
>
> **Q3.Unclear Captions.** Thank you for your suggestion! To make it clearer, we will add more detailed description in the caption and main part of the figure in the modified version.
>
> **Q4.Why suitable and why ImageNet.** Our final goal is to bridge the performance gap between CNNs and ViTs. Our modifications are done to improve the whole architecture of vision transformers, **solving the intrinsic problem of lack of inductive biases in ViTs**. The chosen datasets are the tools to evaluate the effectiveness of our method. Previous works have proposed enough methods on training from scratch on ImageNet but failed to pay attention to much smaller datasets. Our method succeeds on small datasets and it also works well on larger dataset ImageNet, **showing the effectiveness and generalization of our method on various size dataset**. It indeed solves the intrinsic problem of the architecture. We also provide fine-tuning result in the Appendix in Table 1 and 3, further showing the effectiveness of our method.

---

> > ### Author Response · Authors · 2022-08-02
> > **Response 2 to Reviewer b3Vm**
> >
> > Owing to the limited characters of only one response and we want to anwer your questions in details, we have to add another reply!
> >
> > **Q5.Explanation of HI-MHSA**. The design of Head Token together with Head-Interacted MHSA is the core contribution of our paper. It enables different attribute of the object interacted with each other and resulting a general represent. A comprehensive explanation is as follows.
> >
> > Firstly, the HI-MHSA is inspired with and tally with human's visual principle. We human usually find and focus on several discriminative part for recognition, and a convolutional feature channel often corresponds to a certain type of visual pattern. Thus it is reasonable to dividing channels into several groups for re-calibrating.  From the visualization in our main paper and appendix, such as Fig. 3 to Fig. 5 in the Appendix, the attention map of Head Token demonstrates its attention to different part of the object. Some Head Tokens focus on the head of the object, and some other on the main body. By HI-MHSA, the separated representation of the object can be fused into a general one.
> >
> > Secondly, the HI-MHSA is targeted and helpful to Transformer model. As is known, one of the attributes of vanilla MHSA is data-specific. However, this mechanism only models relationship spatial-wise and the multi-head design manually splits the linearly projected tokens into multiple segments, i.e. multi-head, and conduct self-attention within each head. We argue that this kind of design is a must but has it own disadvantage. Softmax function is an essential component in self-attention. It non-linearly generates attention weights from dot-product result of queries and keys. Its characteristic is **Choosing Only One**, which means only few positions have large weight. So multi-head mechanism splits tokens into multiple segments, conducting self-attention in each head, and choosing different position  in each head. **Here we can consider the different segment of the tokens represents different attribute of the object. To make it compatible with multi-head mechanism and enable different attribute of the object can be fused into a general representation, we extract Head Token from different segments of the tokens**. Head Tokens are concatenated with other tokens sequentially, and is processed by vanilla MHSA. With our proposed Head Token design, different head can interact with each other now.
> >
> > **A very interesting discover** is  the visualization of attention maps. As is shown in Fig.7 in the Appendix, in the input layer, all the tokens focus on themselves and head tokens, and in the shallow layers, the patch tokens focus on their neighbors. Further, in the middle layers, all the tokens focus more on the Head Tokens for better representation, and the deep layers draw more attention on prominent patch tokens. This is the visualization of the whole feature extraction process of our method. It raises a good question: **If such characteristic is general in other vision transformer architectures?** Previous works have demonstrated the feature extraction and information exchanging process on spatial aspect. However, they failed to delve into feature exchanging on channel aspect. Here in our paper, we move a step forward in to feature exchanging and integration on channel aspect. Maybe our work is the first to exhibit a potential feature extraction pattern in general vision transformers. And we hope it will inspire more future research.
> >
> > **Q6.BatchNorm/LayerNorm?** Here in our work, BatchNorm is adopted at two positions: SOPE and DAFF. In the following experiments of , we use **(A-B)** to denote normalization choice, where **A is the norm operation in SOPE and B is in DAFF.**  This ablation study is conduct on DHVT-T, training from scratch on CIFAR-100 under the same setting as the main paper. We evaluate the influence from batchsize of 128, 256 and 512. From the following table, we can see  that using BN is indeed sensitive to the batch size, while its performance is always superior to LN. We use serials of convolution operation and so the BN is more compatible. If you want to use LN, you have to reshape the feature and reshape it back for the next convolution.
> > | Method |b128| b256 |b512|
> > |:--:|:--:|:--:|:--:|
> > | BN-BN| 79.69 |80.31|**80.98**|
> > | BN-LN| 78.69 |79.55|79.25|
> > | LN-BN| 79.24 |80.02|80.26
> > | LN-LN| 78.46 |79.09|78.90

---

> > > ### Comment · Reviewer_b3Vm · 2022-08-07
> > > **Post-rebuttal discussion**
> > >
> > > Thanks for the clarification!
> > >
> > > My concerns are partially addressed. I still need precision on some points in order to adjust my final recommendation. The remaining questions are as follows:
> > >
> > > - "We add a shortcut alongside the DWCONV, resulting a data-specific representation of the input data." and ”Though the previous work like CoaT and DaViT try to enhance spatial-wise and channel-wise representation simultaneously, we enhance channel-wise representation in a different way."
> > >
> > >     - As far as I know, the shortcut alongside the DWCONV is also proposed by previous works, like CoaT and DaViT. I highly suggest you state the differences more clearly between the two works and yours in your manuscript.
> > >
> > > - "To make it clearer, we will add more detailed description in the caption and main part of the figure in the modified version."
> > >
> > >     - Seems no update in captions until now.
> > >
> > > - A new question.
> > >     - The computational complexity or the inference time compared with baselines, eg, the origin ViT?

---

> > > > ### Author Response · Authors · 2022-08-08
> > > > **Post-rebuttal Response to Reviewer b3Vm**
> > > >
> > > > Thank you for your further comments! We add more results in this response and we hope this time we can address all your concerns.
> > > >
> > > > **Q1.Updated Version.** We are sorry that we did not update the version before. Now the main paper is updated, following all the suggestions from all the reviewers. Thank you again for the sugeestions.
> > > >
> > > > **Q2. The shortcut alongside.** As is also mentioned by Reviewer 4fHM, the CMT [1] (which we missed before) and Shunted Transformer [2] (which we have cited) have adopted such shortut alongside. These two works came publised in CVPR2022 in March. Meanwhile we also have conducted plenty of experiments of about this shortcut, evaluating its effectiveness on small datasets. So in the final structure of our method, we adopt the shortcut. The difference between our DAFF and the FFN in CMT and Shunted-Trasnformer is that we leverage class token and examine the shortcut alongside DWCONV can also help in this circumstance. Our DAFF can show the effectiveness of shortcut alongside on the other hand. **So combining our work and CMT and Shunted-Transformer together, we can conclude that the shortcut alongside is a generally useful trick, both on small dataset like CIFAR-100 and larger dataset like ImageNet-1K, no matter if the class token is adopted or not.**
> > > >
> > > > [1] Guo J, Han K, Wu H, et al. Cmt: Convolutional neural networks meet vision transformers[C]//Proceedings of the IEEE/CVF Conference on Computer Vision and Pattern Recognition. 2022: 12175-12185.
> > > >
> > > > [2] Ren S, Zhou D, He S, et al. Shunted Self-Attention via Multi-Scale Token Aggregation[C]//Proceedings of the IEEE/CVF Conference on Computer Vision and Pattern Recognition. 2022: 10853-10862.
> > > >
> > > >
> > > > **Q3. Compared with CoaT and DaViT.** Thank you for suggestion. CoaT and DaViT investigate channel-wise representation through conducting self-attention on channel dimension, while we enhance channel-wise representation by dynamically aggregating feature from patch tokens to enhance class token channel-wise and involve channel group-wise head tokens into vanilla self-attention. We update the main paper and the dicussion is placed at L135-L138 now.
> > > >
> > > > **Q4. Computational Complexity of baselines.** We measure the computational complexity on CIFAR-100 under the same circumstance as the Section 2.2 in our Supplementary Materials. And for detailed comparison, we show two digits after the decimal point. Note that we use 4 heads and 8 heads in our DHVT-T and DHVT-S, and for fair comparison, the baseline ViT-Tiny (DeiT-Tiny) and ViT-S (DeiT-S) also adopt 4 and 8 heads respectively.  We further exhibits the results of original ViT with different patch size on CIFAR-100. And during rebuttal period, we further implement CvT-13 on CIFAR-100 with patch size of 2 and 4. So all the results are summarized as follows. Compared with vanilla ViT and CvT, our method reaches much higher performance while the computational complexity increase is reasonable.
> > > >
> > > > From Row 1 and 3, it is not surprising that using smaller patch size will decrease the performance of vanilla ViT. Because ViT is hard to learn spatial relevance under insufficient training data. Using patch size of 4 means the nearby 4x4 pixels are fused into one token, and such token maintains part of the neighboring information. **Decreasing the patch size to 2 further intensifies the non-overlapping problem** and thus decreases the performance of original ViT. And **when scaling up**, from Row 1 to 2, the performance of vanilla ViT also drops, demonstrating that larger amount of parameters in the model are hard to adjust well under insufficient training data. Compared with ViT-S with patch size of 2 on Row 4, the ViT-T with patch size of 4 on Row 1 is superior in the number of parameters, the computational complexity and performance.
> > > >
> > > > While in our DHVT, from Row 7 to 10, using smaller patch size and scaling up can bring higher performance consistently. Smaller patch size is helpful to better investigate local patterns and thus the performance increases. And the scalibility of our DHVT is also promising because the channel representation is enhanced and the parameters are easier to adjust.
> > > >
> > > >
> > > > |Row| Method |Img Size| Patch Size |#Params| GFLOPs | ACC |
> > > > |:--:|:--:|:--:|:--:|:--:|:--:|:--:|
> > > > |1| ViT-T | 32 | 4 |5.38M |0.36| 67.59|
> > > > |2| ViT-S | 32 | 4 |21.38M| 1.42 | 62.05 |
> > > > |3| ViT-T | 32 | 2 |5.41M| 1.68 | 65.86 |
> > > > |4| ViT-S | 32 | 2 |21.44M| 6.08 | 66.04 |
> > > > |5| CvT-13 | 32 | 4 |19.59M| 1.11 | 79.24|
> > > > |6| CvT-13 | 32 | 2 |19.59M| 4.53 | 81.81 |
> > > > |7| DHVT-T | 32 | 4 |6.01M| 0.39 | 80.93 |
> > > > |8| DHVT-S | 32 | 4 |23.43M| 1.54 | 82.91 |
> > > > |9| DHVT-T | 32 | 2 |5.84M| 1.74 | 83.54 |
> > > > |10| DHVT-S | 32 | 2 |22.77M| 6.26 | 85.68 |

---

> > > > > ### Comment · Reviewer_b3Vm · 2022-08-08
> > > > > **Thanks for the update**
> > > > >
> > > > > Thanks for the update. I increase my vote to 5.

---

> > > > > > ### Author Response · Authors · 2022-08-09
> > > > > > **Thanks for reviews and discussion**
> > > > > >
> > > > > > Thanks again for your time, your detailed and insightful comments and kindess! It is a good trip of us these days and your suggestions greatly improve our work, making it more solid. Best wishes.

---

### Meta-Review · Area_Chair_NUoP · 2022-08-20

**Recommendation:** Accept
**Confidence:** Less certain

**Metareview:**

Authors introduce 3 modifications to ViT architecture to introduce additional inductive biases to improve performance in low-data scenarios:
- SOPE: Sequential Overlapping Patch Embedding -- essentially convolutions before partitioning the image into patches.
- DAFF: Dynamic Aggregation Feed Forward -- a DWCONV operation is applied to tokens after a FC layer increases the channel dimension. The new tokens are average pooled, input to additional FC layers, and then are used to scale the CLS token.
- HI-MHSA: Head-Interacted Multi-Head Self-Attention -- this approach's name is confusing. This does not change the heads in MHSA. Rather a new mechanism is introduced prior to MHSA to introduce new tokens where each new token is derived from a different partition of the original channel dimensions.  AC recommends authors use a different name. For example, "Intra-Channel Modeling (ICM) MHSA" or something would be more clear.

Performance is evaluated on CIFAR-100, DomainNet subsets, and ImageNet 1K

Pros:
- [R/AC] The topic is important to the community.
- [R/AC] The paper is well written and clear.
- [R/AC] The authors present improved performance versus other recent SOTA hybrid model designs (during rebuttal phase, though missing from original work -- should be added to paper).

Cons:
- [R/AC] The evaluation could be significantly improved. For example, more training experiments on undersampled version of more datasets, with comparisons to other SOTA methods.
- [R/AC] The design is complicated and the motivation isn't always clear.
- [R] Novelty of the components implemented is low.
- [R] Concerns over use of BN as opposed to LN. Authors have provided ablation experiments to demonstrate that BN improves performance of their model over LN. These ablations should be included in the manuscript.
- [R/AC] Concerns over lack of comparison to other SOTA methods that mix convolutions with transformers, such as CvT. Authors have provided additional experiment tables that compare against CvT. These tables should be included in the manuscript in a consistent manner (showing number of parameters and FLOPS).
- [R/AC] Authors do not include FLOPS in their experiment tables. Please ensure all tables report number of parameters and FLOPS for all models explored. There are python packages to help with computing this, such as "flopth".
- [AC] Some spelling and grammatical mistakes. Please spell check the manuscript.

Overall Recommendation: Reviews lean toward acceptance, but marginally so. Given that the authors have provided more comparisons against recent relevant SOTA methods, and that the reviewers (including expert in the field) lean toward accept, the AC opinion is that this manuscript can be published and provides some valuable knowledge to the community. There are ways in which the paper can still be improved before publication, such as inclusion of additional evaluation datasets.

AC Rating: Borderline Accept


**Award:**

No

---

### Decision · Program_Chairs · 2022-09-14

Accept